# Parallel Scaling Law for Language Models

**Mouxiang Chen**[1,2]*, **Binyuan Hui**[2,†], **Zeyu Cui**[2], **Jiaxi Yang**[2],
**Dayiheng Liu**[2], **Jianling Sun**[1], **Junyang Lin**[2], **Zhongxin Liu**[1,†]
[1]Zhejiang University, [2]Qwen Team, Alibaba Group
{chenmx,liu_zx}@zju.edu.cn, binyuan.hby@alibaba-inc.com

## Abstract

It is commonly believed that scaling language models should commit a significant space or time cost, by increasing the parameters (*parameter scaling*) or output tokens (*inference-time scaling*). We introduce another and more inference-efficient scaling paradigm: increasing the model's *parallel computation* during both training and inference time. We apply $P$ diverse and learnable transformations to the input, execute forward passes of the model in parallel, and dynamically aggregate the $P$ outputs. This method, namely *parallel scaling* (PARSCALE), scales parallel computation by reusing existing parameters and can be applied to any model structure, optimization procedure, data, or task. We theoretically propose a new scaling law and validate it through large-scale pre-training, which shows that a model with $P$ parallel streams is similar to scaling the parameters by $\mathcal{O}(\log P)$ while showing superior inference efficiency. For example, PARSCALE can use up to $22\times$ less memory increase and $6\times$ less latency increase compared to parameter scaling that achieves the same performance improvement. It can also recycle an off-the-shelf pre-trained model into a parallelly scaled one by post-training on a small amount of tokens, further reducing the training budget. The new scaling law we discovered potentially facilitates the deployment of more powerful models in low-resource scenarios, and provides an alternative perspective for the role of computation in machine learning. Our code and 67 trained model checkpoints are publicly available at `https://github.com/QwenLM/ParScale` and `https://huggingface.co/ParScale`.

## 1 Introduction

Recent years have witnessed the rapid scaling of large language models (LLMs) [10, 64, 55, 72] to narrow the gap towards Artificial General Intelligence (AGI). Mainstream efforts focus on *parameter scaling* [40], a practice that requires substantial space overhead. For example, DeepSeek-V3 [51] scales the model size up to 672B parameters, which imposes prohibitive memory requirements for edge deployment. More recently, researchers have explored *inference-time scaling* [65] to enhance the reasoning capability by scaling the number of generated reasoning tokens. However, inference-time scaling is limited to certain scenarios and necessitates specialized training data [20, 68], and typically imposes significant time costs. For example, Chen et al. [13] find that the most powerful models can generate up to 900 reasoning tokens for trivial problems like "2+3=?". This motivates the question: **Is there a universal and efficient scaling approach that avoids excessive space and time costs?**

We draw inspiration from classifier-free guidance (CFG) [34], a widely used trick during the inference phase of diffusion models [35], with similar concepts also developed in the NLP community [75, 50]. Unlike traditional methods that use a single forward pass, CFG utilizes *two* forward passes during inference: it first performs a normal forward pass to obtain the first stream of output, then perturbs

---

*Work is partially done during internship in Qwen Team, Alibaba Group.   †Corresponding authors.

39th Conference on Neural Information Processing Systems (NeurIPS 2025).

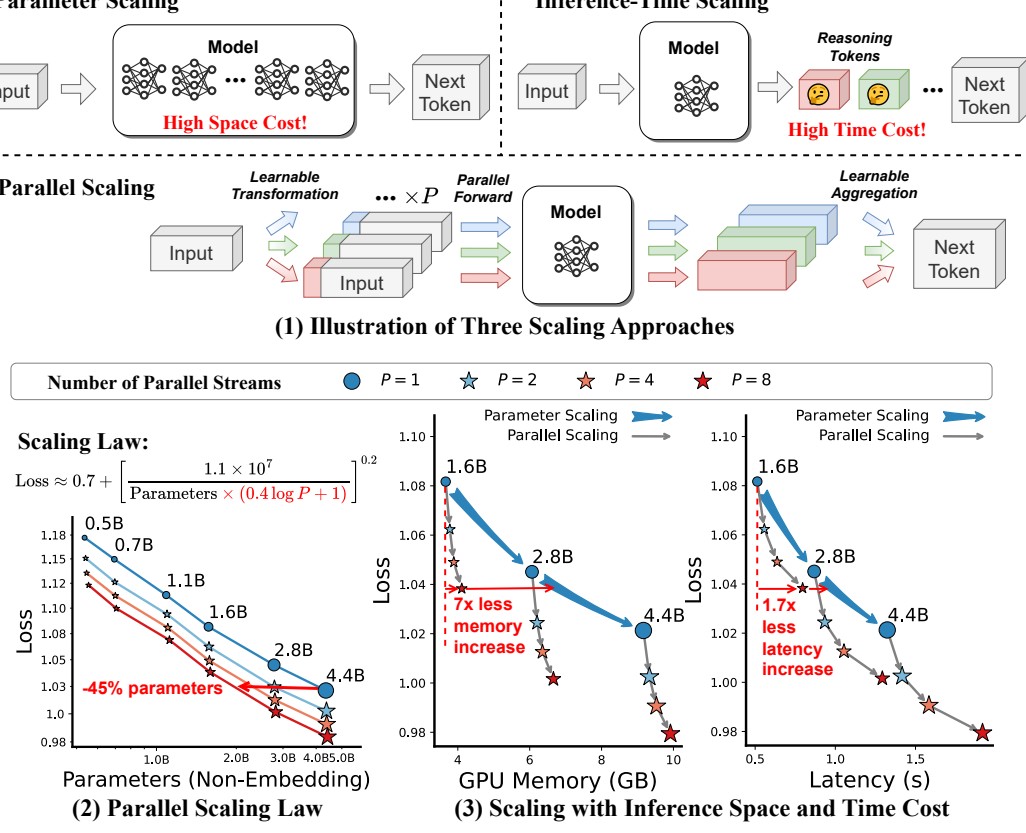

Figure 1: (1) Illustrations of our proposed parallel scaling (PARSCALE). (2) Parallel scaling laws for pre-training models on 42B tokens from Stack-V2 (Python subset). (3) Loss scaling curve with inference cost. Results are averaged from batch size $\in \{1, 2, 4, 8\}$ and input + output tokens $\in \{128, 256, 512, 1024\}$.

the input (e.g., by discarding conditions in the input) to get a second stream of output. The two streams are aggregated based on predetermined contrastive rules, yielding superior performance over single-pass outputs. Despite its widespread use, the theoretical guarantee of CFG remains an open question. In this paper, we hypothesize that the effectiveness of CFG lies in its *double computation*. We further propose the following hypothesis:

**Hypothesis 1.** *Scaling parallel computation (while maintaining the nearly constant parameters) enhances the model's capability, with similar effects as scaling parameters.*

We propose a proof-of-concept scaling approach called *parallel scaling* (PARSCALE) to validate this hypothesis on language models. The core idea is to increase the number of parallel streams while making the input transformation and output aggregation learnable. We propose appending $P$ different learnable prefixes to the input and feeding them in parallel into the model. These $P$ outputs are then aggregated into a single output using a dynamic weighted sum, as shown in Figure 1(1). This method efficiently scales parallel computation during *both* training and inference time by recycling existing parameters, which applies to various training algorithms, data, and tasks.

Our preliminary theoretical analysis suggests that the loss of PARSCALE may follow a power law similar to the Chinchilla scaling law [36]. We then carry out large-scale pre-training experiments on the Stack-V2 [58] and Pile [26] corpus, by ranging $P$ from 1 to 8 and model parameters from 500M to 4.4B. We use the results to fit a new *parallel scaling law* that generalizes the Chinchilla scaling law, as depicted in Figure 1(2). It shows that **parallelizing into P streams equates to scaling the model parameters by $\mathcal{O}(\log P)$**. Results on comprehensive tasks corroborate this conclusion. Unlike parameter scaling, PARSCALE introduces negligible parameters and increases only a little space overhead. It also leverages GPU-friendly parallel computation, shifting the memory bottleneck

Table 1: Comparisons of mainstream LLM scaling strategies. We subdivide parameter scaling into traditional **Dense Scaling** and **Mixture-of-Expert (MoE) Scaling** [23] for comparison. **Inference-Time Scaling**: Enhancing the reasoning ability through large-scale reinforcement learning (RL) to scale reasoning tokens during inference.

| Method | Inference Time | Inference Space | Training Cost | Specialized Strategy |
|---|---|---|---|---|
| Dense Scaling | 🙂 Moderate | 😠 High | 😠 Pre-training only | 😄 No |
| MoE Scaling | 😄 Low | 😠 High | 😠 Pre-training only | 😠 Load balancing |
| Inference-Time Scaling | 😠 High | 🙂 Moderate | 😬 Post-training | 😠 RL / reward data |
| Parallel Scaling | 🙂 Moderate | 🙂 Moderate | 😀 Pre- or Post-training | 😄 No |

in LLM decoding to a computational bottleneck and, therefore, does not notably increase latency. For example, for a 1.6B model, when scaling to $P = 8$ using PARSCALE, it uses **22× less memory increase** and **6× less latency increase** compared to parameter scaling that achieves the same model capacity (batch size = 1, detailed in Section 3.3). Figure 1(3) illustrates that PARSCALE offers superior inference efficiency.

Furthermore, we show that the high training cost of PARSCALE can be reduced by a two-stage approach: the first stage employs traditional training with most of the training data, and PARSCALE is applied only in the second stage with a small number of tokens. Based on this, we train 1.8B models with various $P$ and scale the training data to 1T tokens. The results of 21 downstream benchmarks indicate the efficacy of this strategy. For example, when scaling to $P = 8$, it yields a 34% relative improvement for GSM8K and 23% relative improvement for MMLU using exactly the same training data. We also implement PARSCALE on an off-the-shelf model, Qwen-2.5 [70], and demonstrate that PARSCALE is effective in both full and parameter-efficient fine-tuning settings. This also shows the viability of **dynamic parallel scaling**, which allows flexible adjustment of $P$ during deployment while freezing the backbone weights, to fit different application scenerios.

Table 1 compares PARSCALE with other mainstream scaling strategies. Beyond introducing an efficient scaling approach for language models, our research also tries to address a more fundamental question in machine learning: **Is a model's capacity determined by the parameters or by the computation, and what is their individual contribution?** Traditional machine learning models typically scale both parameters and computation simultaneously, making it difficult to determine their contribution ratio. The PARSCALE and the fitted parallel scaling law may offer a novel and quantitative perspective on this problem.

We posit that large computing can foster the emergence of large intelligence. We hope our work can inspire more ways to scaling computing towards AGI and provide insights for other areas of machine learning. Our key findings in this paper can be summarized as follows:

- Scaling $P$ times of parallel computation is similar to scaling parameters by a ratio of $\mathcal{O}(\log P)$, and larger models reap greater benefits from PARSCALE.

- Reasoning-intensive tasks (e.g., coding or math) benefit more from PARSCALE, which suggests that scaling computation can effectively push the boundary of reasoning.

- PARSCALE offers superior inference efficiency compared to parameter scaling due to the effective use of memory, particularly suitable for low-resource edge deployment.

- The training cost of PARSCALE can be significantly alleviated through a two-stage training strategy.

- PARSCALE remains effective with frozen main parameters for different $P$. This illustrates the potential of dynamic parallel scaling: switching $P$ to adapt model capabilities during inference.

## 2 Background and Methodology

**Classifier-Free Guidance (CFG)**  CFG [34] has become a *de facto* inference-time trick in diffusion models [35], with similar concepts also developed in NLP [75, 50]. At a high level, these lines of work can be summarized as follows: given an input $x \in \mathbb{R}^{d_i}$ and a trained model $f_\theta : \mathbb{R}^{d_i} \rightarrow \mathbb{R}^{d_o}$, where $\theta$ is the parameter and $d_i, d_o$ are dimensions, we transform $x$ into a "bad" version $x'$ based on some heuristic rules (e.g., removing conditions), obtaining two parallel outputs $f_\theta(x)$ and $f_\theta(x')$.

The final output $g_\theta(x)$ is aggregated based on the following rule:

$$g_\theta(x) = f_\theta(x) + w\left(f_\theta(x) - f_\theta(x')\right). \tag{1}$$

Here, $w > 0$ is a pre-set hyperparameter. Intuitively, Equation (1) can be seen as starting from a "good" prediction and moving $w$ steps in the direction away from a "bad" prediction. Existing research shows that $g_\theta(x)$ can perform better than the vanilla $f_\theta(x)$ in practice [73].

**Motivation**  In Equation (1), $x'$ is simply a degraded version of $x$, suggesting that $g_\theta(x)$ does not gain more useful information than $f_\theta(x)$. This raises the question: **why is $f_\theta(x)$ unable to learn the capability of $g_\theta(x)$ during training, despite both having the same parameters?** We hypothesize that the fundamental reason lies in $g_\theta(x)$ *having twice the computation as* $f_\theta(x)$. This inspires us to further expand Equation (1) into the following form:

$$g_\theta(x) = w_1 f_\theta(x_1) + w_2 f_\theta(x_2) + \cdots + w_P f_\theta(x_P), \tag{2}$$

where $P$ denotes the number of parallel streams. $x_1, \cdots, x_P$ are $P$ distinct transformations of $x$, and $w_1, \cdots, w_P$ are aggregation weights. We term Equation (2) as a *parallel scaling* (PARSCALE) of the model $f_\theta$ with $P$ streams. This scaling strategy does not require changing the structure of $f_\theta$ and training data. In this paper, we focus on Transformer language models [86, 10], and regard the stacked Transformer layers as $f_\theta(\cdot)$.

**Implementation Details and Pivot Experiments**  We apply Equation (2) in both training and inference time, and perform a series of pivot experiments to determine the best input transformation and output aggregation strategies (refer to Appendix A). The findings revealed that **variations in these strategies minimally affect model performance; the significant factor is the number of computations (i.e., $P$)**. Finally, for input transformation, we employ prefix tuning [49] as the input transformation, which is equivalent to using different KV-caches to distinguish different streams. For output aggregation, we employ a dynamic weighted average approach, utilizing an MLP to convert outputs from multiple streams into aggregation weights. This increases about 0.2% additional parameters for each stream.

## 3 Parallel Scaling Law

This section focuses on the in-depth comparison of scaling parallel computation with scaling parameters. In Section 3.1, we theoretically demonstrate that parallel scaling is equivalent to increasing parameters by a certain amount. In Section 3.2, we validate this with a practical scaling law through large-scale experiments. Finally, in Section 3.3, we analyze latency and memory usage during inference to show that parallel scaling is more efficient.

### 3.1 Theoretical Analysis: Can PARSCALE Achieve Similar Effects as Parameter Scaling?

From another perspective, PARSCALE can be seen as an *ensemble* of multiple different next token predictions, despite the ensemble components sharing most of the parameters. Existing theory in literature finds that the ensembling performance depends on the diversity of different components [8, 57]. In this section, we further validate this finding by theoretically proposing a new scaling law that generalizes existing language model scaling laws, and demonstrate that PARSCALE can achieve similar effects as parameter scaling.

We consider a special case that $w_1 = w_2 = \cdots = 1/P$ to simplify our analysis. This is a degraded version of PARSCALE, therefore, we can expect that the full version of PARSCALE is at least not worse than the theoretical results we can obtain (See Appendix A for further numeric comparison). Let $\hat{p}_i(\cdot \mid x) = f_\theta(x_i)$ denote the next token distribution for the input sequence $x$ predicted by the $i$-th stream. Based on Equation (2), the final prediction $\hat{p}(\cdot \mid x) = g_\theta(x)$ is the average across $\hat{p}_i$, i.e., $\hat{p}(\cdot \mid x) = 1/P \sum_i \hat{p}_i(\cdot \mid x)$. Chinchilla [36] proposes that the loss $\mathcal{L}$ of a language model with $N$ parameters is a function of $N$ after convergence. We assume the prediction of each stream adheres to the Chinchilla scaling law, as follows:

**Lemma 3.1** (Chinchilla Scaling Law [36])**.** *The language model cross-entropy loss $\mathcal{L}_i$ for the $i$-th stream prediction (with $N$ parameters) when convergence is:*

$$\mathcal{L}_i = \left(\frac{A}{N}\right)^\alpha + E, \quad 1 \le i \le P, \tag{3}$$

*where $\{A, E, \alpha\}$ are some positive constants. $E$ is the entropy of natural text.*[2]

Based on Lemma 3.1, we theoretically derive that after aggregating $P$ streams, the prediction follows a new type of scaling law, as follows:

**Proposition 1** (Theoretical Formula for Parallel Scaling Law). *The loss $\mathcal{L}$ for PARSCALE (with $P$ streams and $N$ parameters) is*

$$\mathcal{L} = \left( \frac{A}{N \cdot P^{1/\alpha} \cdot \text{DIVERSITY}} \right)^{\alpha} + E. \tag{4}$$

*We define* DIVERSITY *as:*

$$\text{DIVERSITY} = [(P-1)\rho + 1]^{-1/\alpha},$$

*where $\rho$ is the **correlation coefficient** between random variables $\Delta p_i$ and $\Delta p_j$ ($i \neq j$), and $\Delta p_i$ is the **relative residuals** for the $i$-th stream prediction, i.e., $\Delta p_i = [\hat{p}_i(y|x) - p(y|x)]/p(y|x)$. $p(y \mid x)$ is the real next token probability.*

Proof for Proposition 1 is elaborated in Appendix B. From it, we can observe two key insights:

1. When $\rho = 1$, predictions across different streams are identical, at which point we can validate that Equation (4) degenerates into Equation (3). Random initialization on a small number of parameters introduced (i.e., prefix embeddings) is sufficient to avoid this situation in our experiments, likely due to the impact being magnified by the extensive computation of LLMs.

2. When $\rho \neq 1$, $\mathcal{L}$ is inversely correlated to $P$. Notably, when $\rho = 0$, residuals are independent between streams and the training loss exhibits a *power-law* relationship with $P$ (i.e., $\mathcal{L} \propto P^{-1}$). This aligns with findings in Lobacheva et al. [57]. When $\rho$ is negative, the loss can further decrease and approach zero. This somewhat demystifies the effectiveness of CFG: by widening the gap between "good" input $x$ and "bad" input $x'$, we force the model to "think" from two distinct perspectives, which can increase the diversity between the two outputs.

Despite the difficulty in further modeling $\rho$, Proposition 1 suggests that **scaling $P$ times of parallel computation is equivalent to scaling the model parameter count, by a factor of $\left(P^{1/\alpha} \cdot \text{DIVERSITY}\right)$**. This motivates us to go further, by empirically fitting a practical parallel scaling law to validate Hypothesis 1.

### 3.2 Practical Parallel Scaling Laws

**Experiment Setup**    To fit a parallel scaling law in practice, we pre-train Transformer language models with the Qwen-2.5 dense architecture and tokenizer [70] *from scratch* on the open-source corpus. We primarily focus on the relationship between parallel scaling and parameter scaling. Therefore, we fix the training data size at 42 billion tokens without data repeat[3]. We introduce the results for more training tokens in the next section, and leave the impact of data scale on the scaling law for future work. Most of our settings follow existing works [63], detailed in Appendix C.

Our pre-training is conducted on two widely utilized datasets: Stack-V2 (Python subset) [58] and Pile [26]. Pile serves as a general corpus aimed at enhancing common sense and memorization skills, while Stack-V2 focuses on code comprehension and reasoning skills. Analyzing PARSCALE across these contexts can assess how parameters and computations contribute to different skills.

**Parametric Fitting**    We plot the results in Figure 2, where each point represents the loss of a training run, detailed in Appendix F. We observe that increasing $P$ yields benefits following a logarithmic trend. Similar gains are seen when raising $P$ from 1 to 2, 2 to 4, and 4 to 8. Thus, we preliminarily try the following form:

$$\mathcal{L} = \left( \frac{A}{N \cdot (k \log P + 1)} \right)^{\alpha} + E, \tag{5}$$

---

[2]Chinchilla also considers the limited training steps. In this paper, we focus on the impact of computation and parameters on model capacity and assumes that the model has been trained to convergence.

[3]In Appendix D, we test PARSCALE with repeated data on a small-scale corpus, OpenWebText [28], and found that **PARSCALE helps mitigate the overfitting when data is limited**. We leave further exploration on the data-constrained parallel scaling law for future work.

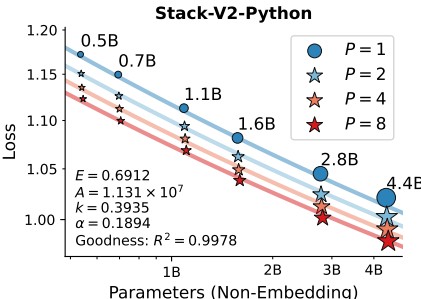
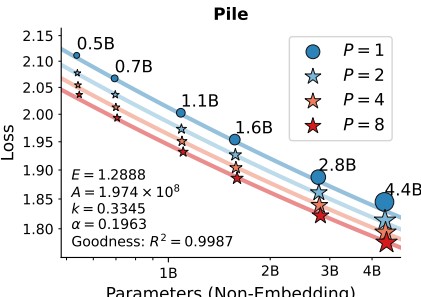

Figure 2: Loss of LLMs scaled on parameters and number of parallel streams $P$ trained on 42B tokens. Each point depicts the loss from a training run. The fitted scaling law curve from Equation (5) is displayed, with annotated fitted parameters $(E, A, k, \alpha)$ and the goodness of fit $R^2$.

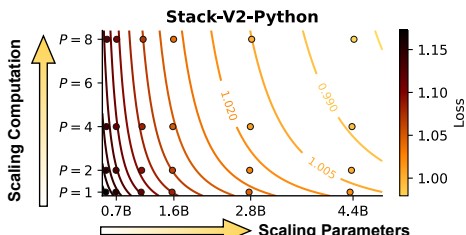
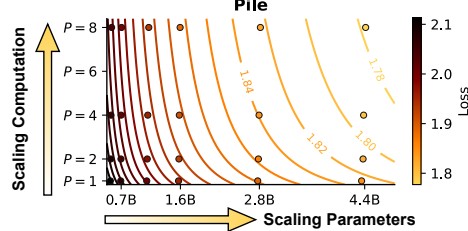

Figure 3: Predicted loss contours for PARSCALE. Each contour line indicates a combination of (parameter, $P$) with similar performance.

where we assume that $P^{1/\alpha} \cdot \text{DIVERSITY} = k \log P + 1$ in Equation (4) based on the finding of the logarithmic trend. $(A, k, \alpha, E)$ are parameters to fit, and we use the natural logarithm (base $e$). We follow the fitting procedure from [36, 63], detailed in Appendix E.

Figure 2 illustrates the parallel scaling law fitted for two training datasets. It shows a high goodness of fit ($R^2$ up to 0.998), validating the effectiveness of Equation (5). Notably, we can observe that the $k$ value for Stack-V2 (0.39) is higher than for Pile (0.33). Recall that $k$ reflects the benefits of increased parallel computation. Since Stack-V2 emphasizes coding and reasoning abilities while Pile emphasizes memorization capacity, we propose an intuitive conjecture that **model parameters mainly impact the memorization skills, while computation mainly impacts the reasoning skills**. This aligns with recent findings on inference-time scaling [27]. Unlike those studies, we further *quantitatively* assess the ratio of contribution to model performance between parameters and computation through our proposed scaling laws.

Recall that Equation (5) implies scaling $P$ equates to increasing parameters by $\mathcal{O}(N \log P)$. It suggests that **models with more parameters benefit more from PARSCALE**. Figure 3 more intuitively displays the influence of computation and parameters on model capacity. As model parameters increase, the loss contours flatten, showing greater benefits from increasing computation.

**Downstream Performance** Tables 2 and 3 illustrate the average performance on downstream tasks (coding tasks for Stack-V2-Python and general tasks for Pile) after pre-training, with comprehensive results in Appendix G. It shows that increasing the number of parallel streams $P$ consistently boosts performance, which confirms that PARSCALE is able to enhance the model capabilities and similar to scale the parameters. Notably, PARSCALE offers more substantial improvements for coding tasks compared to general tasks. For example, as shown in Table 2, the coding ability of the 1.6B model for $P = 8$ aligns with the 4.4B model, while Table 3 indicates that such setting performs comparably to the 2.8B model on general tasks that focusing on common-sense memorization.

### 3.3 Inference Cost Analysis

We further compare the inference efficiency between parallel scaling and parameter scaling at equivalent performance levels. Although some work uses FLOPS to measure the inference cost

Table 2: Average performance (%) on two code generation tasks, HumanEval(+) and MBPP(+), after pre-training on the Stack-V2-Python dataset.

| $N$ | 0.5B | 0.7B | 1.1B | 1.6B | 2.8B | 4.4B |
|---|---|---|---|---|---|---|
| $P = 1$ | 26.7 | 28.4 | 31.6 | 33.9 | 36.9 | 39.2 |
| $P = 2$ | 30.3 | 32.4 | 33.6 | 37.4 | 39.4 | 42.6 |
| $P = 4$ | 30.1 | 32.5 | 34.1 | 37.6 | 40.7 | 42.6 |
| $P = 8$ | 32.3 | 34.0 | 37.2 | 39.1 | 42.1 | 45.4 |

Table 3: Average performance (%) on six general lm-evaluation-harness tasks after pre-training on the Pile dataset.

| $N$ | 0.5B | 0.7B | 1.1B | 1.6B | 2.8B | 4.4B |
|---|---|---|---|---|---|---|
| $P = 1$ | 49.1 | 50.6 | 52.1 | 53.1 | 55.2 | 57.2 |
| $P = 2$ | 49.9 | 51.0 | 52.4 | 54.4 | 57.0 | 58.5 |
| $P = 4$ | 50.6 | 51.8 | 53.3 | 55.0 | 57.8 | 59.1 |
| $P = 8$ | 50.7 | 51.8 | 54.2 | 55.7 | 58.1 | 59.6 |

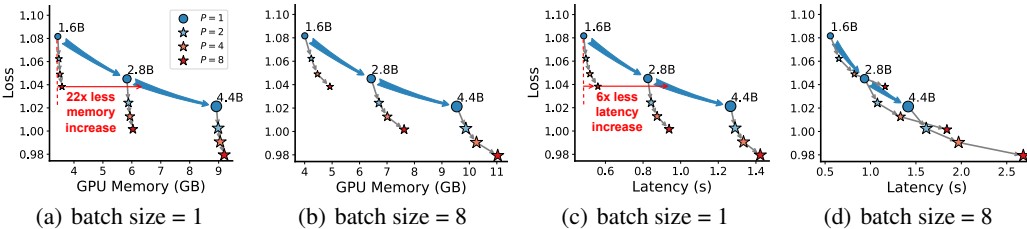

(a) batch size = 1    (b) batch size = 8    (c) batch size = 1    (d) batch size = 8

Figure 4: Loss scales on the inference space-time cost, with three parameters (1.6B, 2.8B, and 4.4B) and two batch sizes. Results are averaged from input / output tokens $\in \{64, 128, 256, 512\}$. Blue arrows indicate parameter scaling; gray arrows represent parallel scaling.

[36, 76], we argue that this is not an ideal metric. Most Transformer operations are bottlenecked by memory access rather than computation during the decoding stage [39]. Some work (such as flash attention [18]) incurs more FLOPS but achieves lower latency by reducing memory access. Therefore, we use **memory** and **latency** to measure the inference cost, based on the llm-analysis framework [48]. We analyze the inference cost across various inference batch sizes. It is worth mentioning that all models in our experiments feature the same number of layers, differing only in parameter width and parallel streams (detailed in Appendix C). This enables a more fair comparison of the efficiencies.

**Space Cost** Figures 4(a) and 4(b) compare the inference memory usage of two scaling strategies, where we utilize loss on Stack-V2-Python as an indicator for model capacity. It shows that PARSCALE only marginally increases memory usage, even with larger batch sizes. This is because PARSCALE introduces negligible amounts of additional parameters (i.e., prefix tokens and aggregation weights, about 0.2% parameters per stream) and increases KV cache size (expanded by $P$ times with $P$ streams), which generally occupies far less GPU memory than model parameters. As the input batch size increases, the KV cache size also grows; however, PARSCALE continues to demonstrate significantly better memory efficiency compared to parameter scaling. This suggests that **PARSCALE maximizes the utility of memory through parameter reusing, while parameter scaling employs limited computation per parameter and cannot fully exploit computation resources.**

**Time Cost** Figures 4(c) and 4(d) compare the inference time of two scaling strategies. It shows that PARSCALE adds minimal latency at smaller batch sizes since the memory bottlenect is converted tothe computation bottleneck. Given that parallel computation introduced by PARSCALE is friendly to GPUs, it will not significantly raise latency. As batch sizes increase, decoding shifts from a memory to a computation bottleneck, resulting in higher costs for PARSCALE, but it remains more efficient than parameter scaling up to a batch size of 8.

The above analysis indicates that PARSCALE is ideal for low-resource edge devices like smartphones, smart cars, and robots, where queries are typically few and batch sizes are small. Given limited memory resources in these environments, PARSCALE effectively utilizes memory and latency advantages at small batch sizes. When batch size is 1, for a 1.6B model and scaling to P = 8 using PARSCALE, it uses 22× less memory increase and 6× less latency increase compared to parameter scaling that achieves the same performance. **We anticipate that the future LLMs will gradually shift from centralized server deployments to edge deployments with the popularization of artificial intelligence. This suggests the promising potential of PARSCALE in the future.**

Table 4: Performance comparison of the 1.8B models after training on 1T tokens from scratch using the two-stage strategy. We incorporate recent strong baselines (less than 2B parameters) as a comparison to validate that our $P = 1$ baseline is well-trained. The best performance and its comparable performance (within 0.5%) is bolded. Appendix G elaborates the evaluation details.

| | Tokens | Data | Average | | | General | | | | | |
| | | | General | Math | Code | MMLU | WinoGrande | Hellaswag | OBQA | PiQA | ARC |
|---|---|---|---|---|---|---|---|---|---|---|---|
| gemma-3-1B | 2T | Private | 53.4 | 1.9 | 14.9 | 26.4 | 61.4 | 63.0 | 37.8 | 75.6 | 56.2 |
| Llama-3.2-1B | 15T | Private | 54.8 | 4.7 | 30.1 | 30.8 | 62.1 | 65.7 | 39.2 | 75.9 | 55.3 |
| Qwen2.5-1.5B | 18T | Private | 63.6 | 52.3 | 55.8 | 61.0 | 65.6 | 68.0 | 42.6 | 76.6 | 67.9 |
| SmolLM-1.7B | 1T | Public | 57.0 | 6.0 | 37.9 | 29.7 | 61.8 | 67.3 | 42.8 | 77.3 | 63.3 |
| SmolLM2-1.7B | 12T | Public | 63.3 | 24.3 | 41.6 | 50.1 | 68.2 | 73.1 | 42.6 | 78.3 | 67.3 |
| Baseline ($P = 1$) | 1T | Public | 56.0 | 25.5 | 45.6 | 28.5 | 61.9 | 65.0 | 40.6 | 75.2 | 64.8 |
| PARSCALE ($P = 2$) | 1T | Public | 56.2 | 27.1 | 47.4 | 29.0 | 62.4 | 64.7 | 42.0 | 74.9 | 64.3 |
| PARSCALE ($P = 4$) | 1T | Public | 57.2 | 30.0 | 48.6 | 30.0 | 63.4 | 65.9 | 42.0 | 75.6 | **66.3** |
| PARSCALE ($P = 8$) | 1T | Public | **58.6** | **32.8** | **49.9** | **35.1** | **64.9** | **67.0** | 42.6 | 76.1 | 66.0 |

| | Tokens | Data | Math | | | Code | | | | | | | |
| | | | GSM8K | GSM8K +CoT | Minerva Math | HumanEval @1 | @10 | HumanEval+ @1 | @10 | MBPP @1 | @10 | MBPP+ @1 | @10 |
|---|---|---|---|---|---|---|---|---|---|---|---|---|---|
| gemma-3-1B | 2T | Private | 1.8 | 2.3 | 1.5 | 6.7 | 15.9 | 6.1 | 14.6 | 13.0 | 29.1 | 10.8 | 23.0 |
| Llama-3.2-1B | 15T | Private | 5.1 | 7.2 | 1.8 | 16.5 | 27.4 | 14.0 | 25.0 | 33.1 | 54.5 | 27.0 | 43.4 |
| Qwen2.5-1.5B | 18T | Private | 61.7 | 67.2 | 28.1 | 36.0 | 62.8 | 31.1 | 55.5 | 61.9 | 79.6 | 50.8 | 68.5 |
| SmolLM-1.7B | 1T | Public | 6.4 | 8.3 | 3.2 | 20.1 | 35.4 | 15.9 | 32.3 | 40.7 | 66.4 | 34.7 | 57.4 |
| SmolLM2-1.7B | 12T | Public | 30.4 | 30.8 | 11.8 | 23.8 | 44.5 | 18.9 | 37.8 | 45.2 | 68.5 | 36.0 | 57.9 |
| Baseline ($P = 1$) | 1T | Public | 28.7 | 35.9 | 12.0 | 26.8 | 44.5 | 20.7 | 38.4 | 51.6 | 75.9 | 43.9 | 62.7 |
| PARSCALE ($P = 2$) | 1T | Public | 32.6 | 35.6 | 13.0 | 26.2 | 50.0 | 20.1 | 42.1 | 52.9 | 77.0 | 45.0 | 65.6 |
| PARSCALE ($P = 4$) | 1T | Public | 34.7 | 40.8 | 14.5 | 27.4 | 47.6 | 23.8 | 43.9 | 55.3 | 77.0 | 47.1 | **66.7** |
| PARSCALE ($P = 8$) | 1T | Public | **38.4** | **43.7** | **16.4** | **28.7** | **50.6** | **24.4** | **44.5** | **56.3** | **79.1** | **48.1** | 67.2 |

## 4 Scaling Training Data

Due to our limited budget, our previous experiments on scaling laws focus on pre-training with 42 billion tokens. In this section, we will train a 1.8B model (with 1.6B non-embedding parameters) and scale the training data to 1T tokens, to investigate whether PARSCALE is effective for production-level training. We also apply PARSCALE to an off-the-shelf model, Qwen-2.5 [70] (which is pre-trained on 18T tokens), under two settings: continual pre-training and parameter-efficient fine-tuning (PEFT).

### 4.1 Two-Stage Pretraining

While PARSCALE is efficient for the inference stage (as we discuss in Section 3.3), it still introduces about $P$ times of floating-point operations and significantly increases overhead in the computation-intensive training processes. To address this limitation, we propose a two-stage strategy: in the first stage, we use traditional pre-training methods with 1 trillion tokens; in the second stage, we conduct PARSCALE training with 20 billion tokens. Since the second stage accounts for only 2% of the first stage, this strategy can greatly reduce training costs. This two-stage strategy is similar to long-context fine-tuning [22], which also positions the more resource-intensive phase at the end. In this section, we will examine the effectiveness of this strategy. Pre-training setup is elaborated in Appendix C.

**Training Loss** Figure 5(a) demonstrates the loss curve during our two-stage training. At the beginning of the second phase, the loss for $P > 1$ initially exceed those for $P = 1$ due to the introduction of randomly initialized parameters. However, after processing a small amount of data (0.0002T tokens), the model quickly adapts to these newly introduced parameters and remains stable thereafter. This proves that PARSCALE can take effect rapidly with just a little data. We can also find that in the later stages, PARSCALE yields similar logarithmic gains. This aligns with previous scaling law findings, suggesting that our earlier conclusions for from-scratch pre-training — *parallelism with $P$ streams equates to a $\mathcal{O}(N \log(P))$ increase in parameters* — also applies to continued pretraining. Additionally, larger $P$ (such as $P = 8$) can gradually widen the gap compared to smaller $P$ values (such as $P = 4$). This demonstrates that parallel scaling can also benefit from data scaling.

**Downstream Performance** In Table 4, we report the downstream performance of the model after finishing two-stage training, across 7 general tasks, 3 math tasks, and 8 coding tasks. It can be

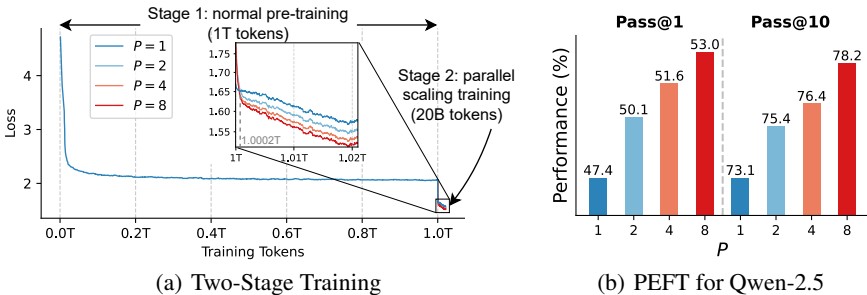

(a) Two-Stage Training
(b) PEFT for Qwen-2.5

Figure 5: (a) Loss for two-stage training, smoothing using an exponential moving average with a weight of 0.95. (b) Code generation performance after fine-tuning on Stack-V2 (Python), averaged from HumanEval(+) and MBPP(+). We only fine-tune the introduced parameters (prefix tokens and aggregation weights), with different $P$ sharing exactly the same Qwen2.5-3B pre-trained weights.

observed that with the increase of $P$, the performance presents an upward trend for most of the benchmarks, which validates the effectiveness of PARSCALE trained on the large dataset. Specifically, when $P$ increases from 1 to 8, PARSCALE improves by 2.6% on general tasks, and by 7.3% and 4.3% on math and code tasks, respectively. It achieves a 10% improvement (34% relative improvement) on GSM8K. This reaffirms that PARSCALE can more effectively address reasoning-intensive tasks. Moreover, in combination with CoT, it achieves about an 8% improvement on GSM8K, suggesting that parallel scaling can be used together with inference-time scaling to achieve better results.

To validate the applicability of PARSCALE to other training stages and data, we use instruction tuning for the base models as detailed in Appendix I.

### 4.2 Applying to the Off-the-Shelf Pre-Trained Model

We further investigate applying PARSCALE to off-the-shelf models under two settings: continual pre-training and parameter-efficient fine-tuning (PEFT). Specifically, we use Pile and Stack-V2 (Python) to continue pre-training the Qwen-2.5 (3B) model. The training settings remain consistent with Appendix C, with the only difference being that we initialize with Qwen2.5-3B weights and adjust the RoPE base to the preset 1,000,000.

The results for full continual pre-training is elaborated in Appendix J. We further utilize PEFT to fine-tune the introduced parameters while freezing the backbone. Figure 5(b) shows that this strategy can still significantly improve downstream code generation performance. Moreover, this demonstrates the promising prospects of *dynamic parallel scaling*: we can deploy the same backbone and flexibly switch between different numbers of parallel streams in various scenarios (e.g., high throughput and low throughput), which enables quick transitions between different levels of model capacities.

## 5 Related Work

Beyond language modeling, our work can be connected to various machine learning domains. First, scaling computation while maintaining parameters is also the core idea of **inference-time scaling**. Second, as previously mentioned, PARSCALE can be viewed as a dynamic and scalable **classifier-free guidance**. Third, our method can be seen as a special case of **model ensemble**. Lastly, the parallel scaling law we explore is a generalization of the existing **language model scaling laws**.

**Inference-Time Scaling**  The notable successes of reasoning models, such as GPT-o1 [65], DeepSeek-R1 [20], QwQ [71], and Kimi k1.5 [41] have heightened interest in inference-time scaling. These lines of work [91, 59, 102] focus on scaling *serial computation* to increase the length of the chain-of-thought [91]. Despite impressiveness, it results in inefficient inference and sometimes exhibits overthinking problems [13, 83].

Other lines of approaches focus on scaling *parallel computation*. Early methods such as beam search [94], self-consistency [90], and majority voting [11] require no additional training. We also provide an experimental comparison with Beam Search and Majority Voting in Appendix H, which shows the

importance of scaling parallel computing during the training stage. Recently, the proposal-verifier paradigm has gained attention, by employing a trained verifier to select from multiple parallel outputs [9, 95, 101]. However, these methods are limited to certain application scenarios (i.e., generation tasks) and specialized training data (i.e., reward signals).

More recently, Geiping et al. [27] introduce training LLMs to reason within the latent space and scale sequential computation, applicable to any application scenarios without needing specialized datasets. However, this method demands significant serial computation scaling (e.g., 64 times the looping) and invasive model modifications, necessitating training from scratch and complicating integration with existing trained LLMs.

**Classifier-Free Guidance**  Classifier-Free Guidance (CFG) stems from Classifier Guided Diffusion [21], which uses an additional classifier to guide image generation using diffusion models [35]. By using the generation model itself as a classifier, CFG [34] further eliminates dependency on the classifier and leverage two forward passes. Similar concepts have emerged in NLP, such as Coherence Boosting [60], PREADD [66], Context-Aware Decoding [78], and Contrastive Decoding [50]. Recently, Sanchez et al. [75] proposed transferring CFG to language models. However, due to constraints of human-designed heuristic rules, these techniques cannot leverage the power of training-time scaling [40] and the performance is limited.

**Model Ensemble**  Model ensemble is a classic research field in machine learning and is also employed in the context of LLMs [14]. In traditional model ensembles, most ensemble components do not share parameters. Some recent work consider setups with partially shared parameters. For example, Monte Carlo dropout [25] employs multiple different random dropouts during the inference phase, while BatchEnsemble [93, 85] and LoRA ensemble [89] use distinct low-rank matrix factorizations for model weights to differentiate different streams (we also experimented with this technique as input transformation in Appendix A). Weight sharing [96, 44] is another line of work, where some weights of a model are shared across different components and participate in multiple computations. However, these works have not explored the scaling law of parallel computation from the perspective of model capacity. As we discuss in Appendix A, we find that the specific differentiation technique had a minimal impact, and the key factor is the scaling in parallel computation.

**Scaling Laws for Language Models**  Many researchers explore the predictable relationships between LLM training performance and various factors under different settings, such as the number of parameters and data [33, 40, 36, 19, 24], data repetition cycles [63, 32], data mixing [97, 69], and fine-tuning [100]. By extending the predictive empirical scaling laws developed from smaller models to larger models, we can significantly reduce exploration costs. Recently, some studies have investigated the scaling effects during inference [76], noting a log-linear relationship between sampling number and performance [9, 80]. But they are limited to certain application scenarios. Our work extends the Chinchilla scaling law [36] by introducing the intrinsic quantitative relationship between parallel scaling and parameter scaling. Existing literature has also identified a power-law relationship between the number of ensembles and loss in model ensemble scaling laws [56], which can be considered a special case of Proposition 1 when $\rho = 0$.

# 6  Conclusions

In this paper, we propose a new type of scaling strategy, PARSCALE, for training LLMs by reusing existing parameters for multiple times to scale the parallel computation. Our theoretical analysis and extensive experiments propose a parallel scaling law, showing that a model with $N$ parameters and $P\times$ parallel computation can be comparable to a model with $\mathcal{O}(N \log P)$ parameters. We scale the training data to 1T tokens to validate PARSCALE in the real-world practice based on the proposed two-stage strategy, and show that PARSCALE remains effective with frozen main parameters for different $P$. We also demonstrate that parallel scaling is more efficient than parameter scaling during the inference time, especially in low-resource edge scenarios. We elaborate the discussion and future work in Appendix K.

## Acknowledgement

This research/project was partially supported by the National Natural Science Foundation of China (No. 62202420) and Zhejiang Provincial Natural Science Foundation of China (No. LZ25F020003). The authors would like to thank Jiquan Wang, Han Fu, Zhiling Luo, and Yusu Hong for their early idea discussions and inspirations, as well as Lingzhi Zhou for the discussions on efficiency analysis.

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

# Appendix

## Contents

# A   Implementation Details and Pivot Experiments

**Input Transformation**   We expect that the transformations applied to the input embedding $x$ can significantly influence the output, which avoids excessively similar outputs across different parallel streams. This can be achieved through the Soft Prompting technique [46]. Specifically, Lester et al. [46] introduced trainable continuous embeddings, known as soft prompts, which are appended to the original sequence of input word embeddings. Freezing the main network while only fine-tuning these soft prompts can be comparable to full fine-tuning. Building on this concept, prefix-tuning [49] incorporates soft prompts into every attention layer of the Transformer model and appends them to the key and value vectors, showing superiority performance to soft prompts.

We utilize prefix-tuning to implement input transformation. To be specific, we first duplicate the input $x$ into $P$ parallel copies, distinguishing them with different prefixes in each attention layer. This can be implemented as using different KV caches for different streams. We found that randomly initializing the prefixes is sufficient to ensure diverse outputs across different streams. We also leverage the prefix reparameterization trick [49, 29], which is theoretically proved effectiveness by Le et al. [45].

As a comparison, we also compared using other parameter-efficient fine-tuning strategy for discriminating the input, including LoRA [37] and BitFit [5]. Notably, LoRA has been also applied to the model ensemble scenario in the literature [89], but only used in the fine-tuning setting while freezing the main parameters in their experiments.

**Output Aggregation**   We found that using dynamic aggregation weights performs better than static ones. Specifically, we concatenate each output together and use an MLP $h : \mathbb{R}^{d_o \times P} \to \mathbb{R}^P$ to convert it into a vector of length $P$ as aggregation weights. The process can be formalized as:

$$w_1, \cdots, w_P \leftarrow \text{Softmax} \left( h \left( \text{Concat}[f_\theta(x_1); \cdots ; f_\theta(x_P)] \right) \right), \tag{6}$$

where Softmax ensures aggregation weights are normalized. It can be seen as dynamically weighting different parallel streams during forward process for each token. We observed that, in the early stages of training, the model may assign nearly all weight to a few streams, leaving others with near-zero weights. It prevents the prefix parameters of these unlucky streams from receiving gradients and updates. This is similar to the load imbalance phenomenon in sparse MoE architectures [23, 77], where most tokens are sometimes assigned to a few experts. To address this, we apply label smoothing [84] to set a non-zero minimum for each weight, formulated as:

$$w_i \leftarrow w_i \times (1 - \epsilon) + \frac{\epsilon}{P}, \tag{7}$$

where $\epsilon$ denotes the smoothing parameter and we set it to $0.1$ in our experiments.

As a comparison, we also compare using Linear layer to aggregate different outputs and directly average the outputs.

**Results**   We trained a 0.5B model on Stack-V2-Python. Table 5 compares the impact of different strategies on final performance. For output aggregation, a dynamic weighted sum with label smoothing proved most effective. The differences between methods for input transformation were minor (around 0.1%), much less than the benefits obtained from changing $P$. Therefore, we opt for the simplest strategy, prefix tuning. Unlike LoRA and BitFit, it requires minimal changes to the model, only necessitating adjustments to the KV-cache.

Table 5: Comparisons of different strategies for input transformations and output aggregation.

| $P$ | Input Transformation | Output Aggregation | Loss ↓ | Rel. Improvement ↑ |
|---|---|---|---|---|
| 1 | - | - | 1.1518 | 0.00% |
| 2 | Prefix (48 tokens) | Dynamic Weighted Sum ($\epsilon = 0.1$) | 1.1276 | 2.10% |
| 2 | Prefix (48 tokens) | Dynamic Weighted Sum ($\epsilon = 0.0$) | 1.1284 | 2.03% |
| 2 | Prefix (48 tokens) | Average | 1.1288 | 2.00% |
| 2 | Prefix (48 tokens) | Linear Layer | 1.1323 | 1.69% |
| 2 | Prefix (48 tokens) + shared KV cache | Dynamic Weighted Sum ($\epsilon = 0.1$) | 1.1282 | 2.05% |
| 2 | Prefix (48 tokens) | Dynamic Weighted Sum ($\epsilon = 0.1$) | 1.1276 | 2.10% |
| 2 | Prefix (96 tokens) | Dynamic Weighted Sum ($\epsilon = 0.1$) | 1.1278 | 2.08% |
| 2 | LoRA ($r = 2$) | Dynamic Weighted Sum ($\epsilon = 0.1$) | 1.1281 | 2.06% |
| 2 | Prefix (48 tokens) + LoRA ($r = 2$) | Dynamic Weighted Sum ($\epsilon = 0.1$) | 1.1263 | 2.21% |
| 2 | Prefix (48 tokens) + LoRA ($r = 2$) + BitFit | Dynamic Weighted Sum ($\epsilon = 0.1$) | 1.1263 | 2.21% |
| 2 | Prefix (48 tokens) + LoRA ($r = 4$) | Dynamic Weighted Sum ($\epsilon = 0.1$) | 1.1260 | 2.24% |
| 4 | Prefix (48 tokens) | Dynamic Weighted Sum ($\epsilon = 0.1$) | 1.1145 | 3.24% |
| 8 | Prefix (48 tokens) | Dynamic Weighted Sum ($\epsilon = 0.1$) | 1.1019 | 4.33% |

# B Proof for Proposition 1

*Proof.* We first decompose the individual loss $\mathcal{L}_i$. Based on the definition of language model loss, we have:

$$
\begin{aligned}
\mathcal{L}_i &= \mathbb{E}_x \sum_{y \in \mathcal{V}} \left[ -p(y \mid x) \log \hat{p}_i(y \mid x) \right] \\
&= \mathbb{E}_x \sum_{y \in \mathcal{V}} -p(y \mid x) \log \{ p(y \mid x) \times (1 + \Delta p_i(y \mid x)) \} \\
&= \underbrace{\mathbb{E}_x \sum_{y \in \mathcal{V}} \left[ -p(y \mid x) \log p(y \mid x) \right]}_{\text{entropy of natural text}} + \underbrace{\mathbb{E}_x \sum_{y \in \mathcal{V}} -p(y \mid x) \log (1 + \Delta p_i(y \mid x))}_{\text{approximation error for the language model}},
\end{aligned}
$$

where $\mathcal{V}$ is the vocabulary. In Chichilla scaling law, the entropy of natural text is $E$ and the approximation error is $(A/N)^\alpha$. Therefore, we have:

$$
\mathbb{E}_x \sum_{y \in \mathcal{V}} -p(y \mid x) \log (1 + \Delta p_i(y \mid x)) = \left( \frac{A}{N} \right)^\alpha. \tag{8}
$$

Based on the Taylor series expansion, $\log(1 + x) = x - \frac{x^2}{2} + \mathcal{O}(x^3)$. Apply this expansion to Equation (8), we have:

$$
\begin{aligned}
\left( \frac{A}{N} \right)^\alpha &= \mathbb{E}_x \sum_{y \in \mathcal{V}} -p(y \mid x) \left[ \Delta p_i(y \mid x) - \frac{\Delta p_i(y \mid x)^2}{2} + \mathcal{O}\left( \Delta p_i(y \mid x)^3 \right) \right] \\
&= \mathbb{E}_x \left[ \sum_{y \in \mathcal{V}} -(\hat{p}_i(y \mid x) - p(y \mid x)) \right] + \mathbb{E}_x \left[ \sum_{y \in \mathcal{V}} p(y \mid x) \frac{\Delta p_i(y \mid x)^2}{2} \right] + \mathcal{O}\left( \Delta p_i(y \mid x)^3 \right) \\
&= \underbrace{\mathbb{E}_x \left[ \sum_{y \in \mathcal{V}} -\hat{p}_i(y \mid x) + \sum_{y \in \mathcal{V}} p(y \mid x) \right]}_{=0} + \mathbb{E}_x \mathbb{E}_{y \mid x} \frac{\Delta p_i(y \mid x)^2}{2} + \mathcal{O}\left( \Delta p_i(y \mid x)^3 \right) \\
&\sim \mathbb{E}_{x,y} \left[ \frac{\Delta p_i(y \mid x)^2}{2} \right], \tag{9}
\end{aligned}
$$

where the higher-order terms $\mathcal{O}\left( \Delta p_i(y \mid x)^3 \right)$ are omitted in the last line. The results suggest that minimizing the approximation loss of a language model is equal to minimizing the mean square error (MSE) of relative residuals. After fitting the data, an MSE estimator is usually assumed to be *unbiased*, meaning that $\mathbb{E}_{x,y} \Delta p_i(y \mid x) = 0$. Here we follow this unbiased assumption to simplify the following derivation.

We next consider the aggregated loss $\mathcal{L}$. Let $\Delta p(y \mid x)$ denote the new relative residual after the aggregation:

$$
\begin{aligned}
\Delta p(y \mid x) &= \frac{\hat{p}(y \mid x) - p(y \mid x)}{p(y \mid x)} \\
&= \frac{\frac{1}{P} \sum_{i=1}^{P} \hat{p}_i(y \mid x) - p(y \mid x)}{p(y \mid x)} \\
&= \frac{1}{P} \sum_{i=1}^{P} \frac{\hat{p}_i(y \mid x) - p(y \mid x)}{p(y \mid x)} \\
&= \frac{1}{P} \sum_{i=1}^{P} \Delta p_i(y \mid x).
\end{aligned}
$$

Let $\rho$ denote the correlation coefficient between any two relative residuals $\Delta p_i(y \mid x)$ and $\Delta p_j(y \mid x)$ for $i \neq j$. Repeating the above process to decomposite the aggregated loss $\mathcal{L}$, we have:

$$
\mathcal{L} = \mathbb{E}_x \underbrace{\sum_{y \in \mathcal{V}} \left[ -p(y \mid x) \log p(y \mid x) \right]}_{\text{entropy of natural text, equal to } E} + \mathbb{E}_x \underbrace{\sum_{y \in \mathcal{V}} -p(y \mid x) \log\left(1 + \Delta p(y \mid x)\right)}_{\text{approximation error}}
$$

$$
\begin{aligned}
&= E + \mathbb{E}_{x,y} \left[ \frac{\Delta p(y \mid x)^2}{2} \right] \\
&= E + \frac{1}{2} \mathbb{E}_{x,y} \left[ \left( \frac{1}{P} \sum_{i=1}^{P} \Delta p_i(y \mid x) \right)^2 \right] \\
&= E + \frac{1}{2P^2} \mathbb{E}_{x,y} \left[ \sum_{i=1}^{P} \Delta p_i^2(y \mid x) + 2 \sum_{i<j} \Delta p_i(y \mid x) \Delta p_j(y \mid x) \right] \\
&= E + \frac{1}{P^2} \left[ \sum_{i=1}^{P} \mathbb{E}_{x,y} \left[ \frac{\Delta p_i^2(y \mid x)}{2} \right] + 2 \sum_{i<j} \mathbb{E}_{x,y} \left[ \frac{\Delta p_i(y \mid x) \Delta p_j(y \mid x)}{2} \right] \right].
\end{aligned}
$$

Based on the Corollary of Chinchilla Scaling Law (Equation (9)), for the first term:

$$
\mathbb{E}_{x,y} \left[ \frac{\Delta p_i(y \mid x)^2}{2} \right] = \left( \frac{A}{N} \right)^\alpha,
$$

for the cross terms:

$$
\mathbb{E}_{x,y} \left[ \frac{\Delta p_i(y \mid x) \Delta p_j(y \mid x)}{2} \right] = \rho \sqrt{\mathbb{E}_{x,y} \left[ \frac{\Delta p_i^2(y \mid x)}{2} \right]} \sqrt{\mathbb{E}_{x,y} \left[ \frac{\Delta p_j^2(y \mid x)}{2} \right]} = \rho \left( \frac{A}{N} \right)^\alpha.
$$

Combining the results, we obtain the desired result:

$$
\begin{aligned}
\mathcal{L} &= E + \frac{1}{P^2} \left[ P \cdot \left( \frac{A}{N} \right)^\alpha + P(P-1) \cdot \left( \frac{A}{N} \right)^\alpha \cdot \rho \right] \\
&= E + \left( \frac{A}{N} \right)^\alpha \left[ \frac{1 + (P-1)\rho}{P} \right] \\
&= E + \left( \frac{A}{N} \right)^\alpha \cdot \left( \frac{1}{P^{1/\alpha}} \right)^\alpha \cdot \left( \frac{1}{[(P-1)\rho + 1]^{-1/\alpha}} \right)^\alpha \\
&= E + \left( \frac{A}{N P^{1/\alpha} [(P-1)\rho + 1]^{-1/\alpha}} \right)^\alpha.
\end{aligned}
$$

$\square$

# C  Training Details

**Setup for Scaling Law Experiments**  Our training is based on Megatron-LM [79]. We use a batch size of 1024 and a sequence length of 2048, resulting in 20K training steps. Models have up to 4.7 billion parameters (with 4.4B non-embedding parameters) and 8 parallel streams. For models with $P > 1$, we incorporate prefix embeddings and aggregation weight, as introduced in Appendix A. No additional parameters are included for $P = 1$ models to maintain alignment with existing architectures. We report the last step training loss using exponential moving average, with a smoothing weight of 0.95.

For other hyperparameters, the learning rate undergoes a linear warm-up over 2,000 steps, reaching a peak of $3 \times 10^{-4}$ before decreasing to a minimum of $1 \times 10^{-5}$ according to a cosine decay schedule. The models are trained using a batch size of 1,024 and sequence length of 2,048, alongside a RoPE base of 10,000 [82]. We utilize bfloat16 precision and the Adam optimizer [42], setting the epsilon to 1e-8, $\beta_1$ to 0.9, and $\beta_2$ to 0.95. All parameters, including backbones and additional ones we've introduced, are initialized with a Gaussian distribution having a standard deviation of 0.02. Furthermore, we maintain a dropout rate of 0, enforce a weight decay rate of 0.1, and clip gradients at 1.0. The hyperparameter choices are mostly adopted from existing research [63, 36, 70].

**Model Architectures**  The model architectures are mostly based on the dense model of Qwen-2.5 [70]. Recent work has indicated that the number of layers still significantly impacts the final performance of smaller models [1, 54]. To eliminate the influence of the number of layers and derive a cleaner scaling law, we utilize the architecture of Qwen-2.5-3B (comprising 36 layers, 16 attention heads, and 2 KV groups) and vary the hidden size / intermediate size within this framework. By keeping the number of layers constant and increasing the parameter width, we can more fairly compare the latency of parallel scaling and parameter. The final model structure is presented in the Table 6.

Table 6: Model architectures.

| $P$ | Parameters (Non-Embedding) | Hidden Size | Intermediate Size |
|---|---|---|---|
| 1 | 535,813,376 | 896 | 4,864 |
| 2 | 538,195,842 | 896 | 4,864 |
| 4 | 540,577,412 | 896 | 4,864 |
| 8 | 545,340,552 | 896 | 4,864 |
| 1 | 693,753,856 | 1,024 | 5,504 |
| 2 | 696,738,818 | 1,024 | 5,504 |
| 4 | 699,722,756 | 1,024 | 5,504 |
| 8 | 705,690,632 | 1,024 | 5,504 |
| 1 | 1,088,376,320 | 1,280 | 6,912 |
| 2 | 1,092,762,882 | 1,280 | 6,912 |
| 4 | 1,097,148,164 | 1,280 | 6,912 |
| 8 | 1,105,918,728 | 1,280 | 6,912 |
| 1 | 1,571,472,384 | 1,536 | 8,320 |
| 2 | 1,577,522,690 | 1,536 | 8,320 |
| 4 | 1,583,571,460 | 1,536 | 8,320 |
| 8 | 1,595,669,000 | 1,536 | 8,320 |
| 1 | 2,774,773,760 | 2,048 | 11,008 |
| 2 | 2,784,937,986 | 2,048 | 11,008 |
| 4 | 2,795,100,164 | 2,048 | 11,008 |
| 8 | 2,815,424,520 | 2,048 | 11,008 |
| 1 | 4,353,203,200 | 2,560 | 13,824 |
| 2 | 4,368,529,922 | 2,560 | 13,824 |
| 4 | 4,383,854,084 | 2,560 | 13,824 |
| 8 | 4,414,502,408 | 2,560 | 13,824 |

**Setup for Two-Stage Training**    We follow Allal et al. [2] and use the Warmup Stable Decay (WSD) learning rate schedule [38, 99]. In the first stage, employing a 2K step warm-up followed by a fixed learning rate of 3e-4. In the second stage, the learning rate is annealed from 3e-4 to 1e-5. The rest of the hyperparameters remain consistent with the previous experiments.

In the first phase, we do not employ the PARSCALE technique. We refer to the recipe proposed by Allal et al. [2] to construct our training data, which consists of 370B general data, 80B mathematics data, and 50B code data. We train the model for two epochs to consume 1T tokens. Among the general text, there are 345B from FineWeb-Edu [67] and 28B from Cosmopedia 2 [4]; the mathematics data includes 80B from FineMath [2]; and the code data comprises 47B from Stack-V2-Python and 4B from Stack-Python-Edu.

In the second phase, we use the trained model from the first phase as the backbone and introduce additional parameters from PARSCALE, which are randomly initialized using a standard deviation of 0.02 (based on the initialization of Qwen-2.5). Following [2], in this phase, we increase the proportion of mathematics and code data, finally including a total of 7B general text data, 7B mathematics data, and 7B Stack-Python-Edu data.

# D    Training Loss for OpenWebText with Repeating Data

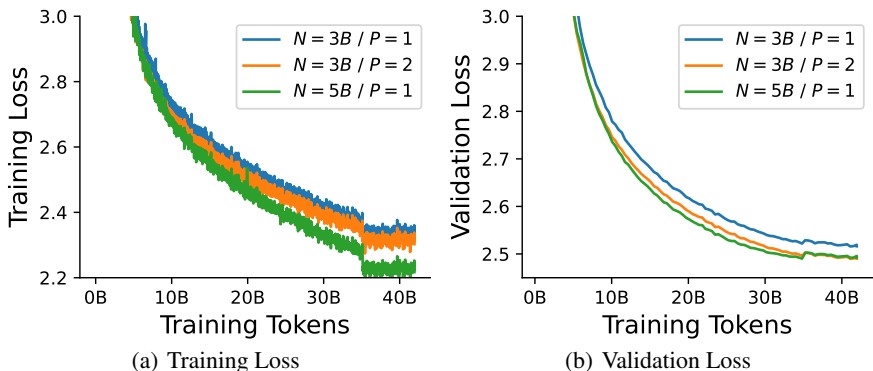

(a) Training Loss          (b) Validation Loss

Figure 6: Loss for training on OpenWebText for repeating several epochs. On the fifth epoch, the validation loss suddenly increases, while **the model with more computations ($N = 3B, P = 2$) shows a stronger resistance to overfitting compared to the model with more parameters ($N = 5B, P = 1$).**

Both Stack-V2-Python and Pile datasets contain more tokens than the total number used in our experiment (42 billion), and therefore previous scaling law experiments did not involve data reuse. Muennighoff et al. [63] noted that the performance of scaling laws tends to change when training data is repeated. In this section, we explore how PARSCALE performs on a smaller dataset, OpenWebText [28], with repeating data.

Figure 6 shows a comparison of training loss and validation loss, with OpenWebText repeated over four cycles. At the transition from the end of the fourth epoch to the beginning of the fifth epoch, we observe a significant decrease in training loss and a notable increase in validation loss, indicating overfitting. This aligns with the optimal number of epochs being four for training language models, as reported by Muennighoff et al. [63]. Comparing parallel scaling and parameter scaling, we observed an intriguing phenomenon: parallel scaling results in a smaller decline in performance when overfitting occurs, while parameter scaling leads to a larger decline. Specifically, by the time overfitting occurs, the validation loss of a 3B parameter model with two-way parallel scaling matched that of a 5 billion parameter model. This suggests that PARSCALE may alleviate the risk of overfitting, possibly due to having fewer parameters. As we increasingly face the depletion of pre-training data [87], further research into the scaling laws of computation in data-constrained scenarios presents a compelling future direction.

# E  Parametric Fitting for the Parallel Scaling Law

To obtain the practical parallel scaling law in Equation (5), based on the 24 runs (i.e., four $P \times$ six $N$) we obtain for each dataset, we follow the strategy from Hoffmann et al. [36] and Muennighoff et al. [63] to use the LBFGS algorithm and Huber loss for curve fitting. This process can be fomulated as:

$$\min_{A,k,E,\alpha} \sum_{\text{run } i} \text{HUBER}_\delta \left( \log \mathcal{L}^i_{\text{pred}}, \log \mathcal{L}^i_{\text{true}} \right),$$

where $\mathcal{L}^i_{\text{true}}$ is the $i$-th observed final loss obtained from experiments and $\mathcal{L}^i_{\text{pred}}$ is the corresponding prediction based on the corresponding observations $\{N, P\}$ and parameters $\{A, k, E, \alpha\}$. We use $\delta = 0.001$ for the Huber loss to avoid overfitting.

Following Muennighoff et al. [63], Hoffmann et al. [36], we utilize the LBFGS algorithm [52] via SciPy [88] to locate local minima of the objective. The initialization grid is defined by: $E \in \{e^{-1}, e^{-0.5}, e^0\}$, $A \in \{e^{-4}, e^{-2}, e^0, e^2, e^4\} \times 10^9$, $\alpha \in \{0, 0.5, 1, 1.5, 2\}$, $k \in \{0.2, 0.4, 0.6, 0.9\}$. All parameters are constrained to be positive. After fitting, the optimal initialization is found to be away from the boundaries of our sweep.

Table 7: Fitting results of the logarithmic scaling law (Equation (5)) for Stack-V2 (Python).

|  |  |
|---|---|
| $A$ | $1.130616 \times 10^7$ |
| $k$ | 0.393463 |
| $E$ | 0.691237 |
| $\alpha$ | 0.189371 |
| Fitting Huber Loss ↓ | $3.677 \times 10^{-5}$ |
| Fitting $R^2$ ↑ | 0.9978 |

Table 8: Fitting results of the logarithmic scaling law (Equation (5)) for Pile.

|  |  |
|---|---|
| $A$ | $1.973520 \times 10^8$ |
| $k$ | 0.334456 |
| $E$ | 1.288766 |
| $\alpha$ | 0.196333 |
| Fitting Huber Loss ↓ | $1.814 \times 10^{-5}$ |
| Fitting $R^2$ ↑ | 0.9987 |

Table 9: Prediction of the logarithmic scaling law (Equation (5)) for Stack-V2 (Python).

| $P$ | Parameters | $\mathcal{L}_{\text{pred}}$ | $\mathcal{L}_{\text{true}}$ | Error |
|---|---|---|---|---|
| 1 | 535,813,376 | 1.1728 | 1.1722 | 0.0006 |
| 1 | 693,753,856 | 1.1498 | 1.1496 | 0.0002 |
| 1 | 1,088,376,320 | 1.1123 | 1.1131 | -0.0008 |
| 1 | 1,571,472,384 | 1.0840 | 1.0817 | 0.0023 |
| 1 | 2,774,773,760 | 1.0439 | 1.0451 | -0.0012 |
| 1 | 4,353,203,200 | 1.0151 | 1.0213 | -0.0062 |
| 2 | 538,195,842 | 1.1509 | 1.1507 | 0.0002 |
| 2 | 696,738,818 | 1.1290 | 1.1262 | 0.0028 |
| 2 | 1,092,762,882 | 1.0932 | 1.0940 | -0.0008 |
| 2 | 1,577,522,690 | 1.0662 | 1.0623 | 0.0039 |
| 2 | 2,784,937,986 | 1.0280 | 1.0244 | 0.0036 |
| 2 | 4,368,529,922 | 1.0005 | 1.0025 | -0.0020 |
| 4 | 540,577,412 | 1.1340 | 1.1354 | -0.0014 |
| 4 | 699,722,756 | 1.1129 | 1.1124 | 0.0005 |
| 4 | 1,097,148,164 | 1.0784 | 1.0808 | -0.0024 |
| 4 | 1,583,571,460 | 1.0524 | 1.0490 | 0.0034 |
| 4 | 2,795,100,164 | 1.0156 | 1.0126 | 0.0030 |
| 4 | 4,383,854,084 | 0.9891 | 0.9906 | -0.0015 |
| 8 | 545,340,552 | 1.1198 | 1.1231 | -0.0033 |
| 8 | 705,690,632 | 1.0994 | 1.0997 | -0.0003 |
| 8 | 1,105,918,728 | 1.0661 | 1.0688 | -0.0027 |
| 8 | 1,595,669,000 | 1.0410 | 1.0383 | 0.0027 |
| 8 | 2,815,424,520 | 1.0053 | 1.0016 | 0.0037 |
| 8 | 4,414,502,408 | 0.9797 | 0.9794 | 0.0003 |

Table 10: Prediction of the logarithmic scaling law (Equation (5)) for Pile.

| $P$ | Parameters | $\mathcal{L}_{\text{pred}}$ | $\mathcal{L}_{\text{true}}$ | Error |
|---|---|---|---|---|
| 1 | 535,813,376 | 2.1107 | 2.1113 | -0.0006 |
| 1 | 693,753,856 | 2.0701 | 2.0671 | 0.0030 |
| 1 | 1,088,376,320 | 2.0039 | 2.0027 | 0.0012 |
| 1 | 1,571,472,384 | 1.9542 | 1.9539 | 0.0003 |
| 1 | 2,774,773,760 | 1.8839 | 1.8876 | -0.0037 |
| 1 | 4,353,203,200 | 1.8335 | 1.8451 | -0.0116 |
| 2 | 538,195,842 | 2.0770 | 2.0772 | -0.0002 |
| 2 | 696,738,818 | 2.0381 | 2.0363 | 0.0018 |
| 2 | 1,092,762,882 | 1.9747 | 1.9730 | 0.0017 |
| 2 | 1,577,522,690 | 1.9270 | 1.9266 | 0.0004 |
| 2 | 2,784,937,986 | 1.8596 | 1.8610 | -0.0014 |
| 2 | 4,368,529,922 | 1.8113 | 1.8137 | -0.0024 |
| 4 | 540,577,412 | 2.0501 | 2.0544 | -0.0043 |
| 4 | 699,722,756 | 2.0125 | 2.0128 | -0.0003 |
| 4 | 1,097,148,164 | 1.9514 | 1.9509 | 0.0005 |
| 4 | 1,583,571,460 | 1.9053 | 1.9040 | 0.0013 |
| 4 | 2,795,100,164 | 1.8402 | 1.8394 | 0.0008 |
| 4 | 4,383,854,084 | 1.7936 | 1.7938 | -0.0002 |
| 8 | 545,340,552 | 2.0272 | 2.0364 | -0.0092 |
| 8 | 705,690,632 | 1.9908 | 1.9933 | -0.0025 |
| 8 | 1,105,918,728 | 1.9315 | 1.9318 | -0.0003 |
| 8 | 1,595,669,000 | 1.8869 | 1.8856 | 0.0013 |
| 8 | 2,815,424,520 | 1.8238 | 1.8218 | 0.0020 |
| 8 | 4,414,502,408 | 1.7785 | 1.7772 | 0.0013 |

Tables 7 and 8 present the fitted parameters and evaluation metrics. Tables 9 and 10 show the prediction results based on the fitted parameters.

Table 11: Fitting results of the theoretical scaling law (Equation (4)) for Stack-V2 (Python).

| | |
|---|---|
| $A$ | $1.187646 \times 10^7$ |
| $\rho$ | 0.891914 |
| $E$ | 0.660016 |
| $\alpha$ | 0.175036 |
| Fitting Huber Loss ↓ | $5.259 \times 10^{-5}$ |
| Fitting $R^2$ ↑ | 0.9959 |

Table 12: Fitting results of the theoretical scaling law (Equation (4)) for Pile.

| | |
|---|---|
| $A$ | $2.150890 \times 10^8$ |
| $\rho$ | 0.899475 |
| $E$ | 1.272178 |
| $\alpha$ | 0.190100 |
| Fitting Huber Loss ↓ | $4.545 \times 10^{-5}$ |
| Fitting $R^2$ ↑ | 0.9968 |

Table 13: Prediction of the theoretical scaling law (Equation (4)) for Stack-V2 (Python).

| $P$ | Parameters | $\mathcal{L}_{\text{pred}}$ | $\mathcal{L}_{\text{true}}$ | Error |
|---|---|---|---|---|
| 1 | 535,813,376 | 1.1734 | 1.1722 | 0.0012 |
| 1 | 693,753,856 | 1.1507 | 1.1496 | 0.0011 |
| 1 | 1,088,376,320 | 1.1135 | 1.1131 | 0.0004 |
| 1 | 1,571,472,384 | 1.0853 | 1.0817 | 0.0036 |
| 1 | 2,774,773,760 | 1.0450 | 1.0451 | -0.0001 |
| 1 | 4,353,203,200 | 1.0158 | 1.0213 | -0.0055 |
| 2 | 538,195,842 | 1.1453 | 1.1507 | -0.0054 |
| 2 | 696,738,818 | 1.1238 | 1.1262 | -0.0024 |
| 2 | 1,092,762,882 | 1.0887 | 1.0940 | -0.0053 |
| 2 | 1,577,522,690 | 1.0620 | 1.0623 | -0.0003 |
| 2 | 2,784,937,986 | 1.0239 | 1.0244 | -0.0005 |
| 2 | 4,368,529,922 | 0.9964 | 1.0025 | -0.0061 |
| 4 | 540,577,412 | 1.1310 | 1.1354 | -0.0044 |
| 4 | 699,722,756 | 1.1102 | 1.1124 | -0.0022 |
| 4 | 1,097,148,164 | 1.0762 | 1.0808 | -0.0046 |
| 4 | 1,583,571,460 | 1.0503 | 1.0490 | 0.0013 |
| 4 | 2,795,100,164 | 1.0133 | 1.0126 | 0.0007 |
| 4 | 4,383,854,084 | 0.9866 | 0.9906 | -0.0040 |
| 8 | 545,340,552 | 1.1234 | 1.1231 | 0.0003 |
| 8 | 705,690,632 | 1.1030 | 1.0997 | 0.0033 |
| 8 | 1,105,918,728 | 1.0695 | 1.0688 | 0.0007 |
| 8 | 1,595,669,000 | 1.0440 | 1.0383 | 0.0057 |
| 8 | 2,815,424,520 | 1.0077 | 1.0016 | 0.0061 |
| 8 | 4,414,502,408 | 0.9814 | 0.9794 | 0.0020 |

Table 14: Prediction of the theoretical scaling law (Equation (4)) for Pile.

| $P$ | Parameters | $\mathcal{L}_{\text{pred}}$ | $\mathcal{L}_{\text{true}}$ | Error |
|---|---|---|---|---|
| 1 | 535,813,376 | 2.1129 | 2.1113 | 0.0016 |
| 1 | 693,753,856 | 2.0726 | 2.0671 | 0.0055 |
| 1 | 1,088,376,320 | 2.0069 | 2.0027 | 0.0042 |
| 1 | 1,571,472,384 | 1.9574 | 1.9539 | 0.0035 |
| 1 | 2,774,773,760 | 1.8872 | 1.8876 | -0.0004 |
| 1 | 4,353,203,200 | 1.8367 | 1.8451 | -0.0084 |
| 2 | 538,195,842 | 2.0700 | 2.0772 | -0.0072 |
| 2 | 696,738,818 | 2.0317 | 2.0363 | -0.0046 |
| 2 | 1,092,762,882 | 1.9695 | 1.9730 | -0.0035 |
| 2 | 1,577,522,690 | 1.9225 | 1.9266 | -0.0041 |
| 2 | 2,784,937,986 | 1.8559 | 1.8610 | -0.0051 |
| 2 | 4,368,529,922 | 1.8080 | 1.8137 | -0.0057 |
| 4 | 540,577,412 | 2.0482 | 2.0544 | -0.0062 |
| 4 | 699,722,756 | 2.0110 | 2.0128 | -0.0018 |
| 4 | 1,097,148,164 | 1.9505 | 1.9509 | -0.0004 |
| 4 | 1,583,571,460 | 1.9048 | 1.9040 | 0.0008 |
| 4 | 2,795,100,164 | 1.8400 | 1.8394 | 0.0006 |
| 4 | 4,383,854,084 | 1.7935 | 1.7938 | -0.0003 |
| 8 | 545,340,552 | 2.0364 | 2.0364 | -0.0000 |
| 8 | 705,690,632 | 1.9998 | 1.9933 | 0.0065 |
| 8 | 1,105,918,728 | 1.9403 | 1.9318 | 0.0085 |
| 8 | 1,595,669,000 | 1.8953 | 1.8856 | 0.0097 |
| 8 | 2,815,424,520 | 1.8315 | 1.8218 | 0.0097 |
| 8 | 4,414,502,408 | 1.7857 | 1.7772 | 0.0085 |

We also test our derived theoretical parallel scaling law (Equation (4)), in which the correlation coefficient $\rho$ is treated as a constant to fit. Tables 11 and 12 present the fitted parameters and evaluation metrics, while Tables 13 and 14 show the fitted results. It is evident that, whether for Stack or Pile, treating $\rho$ as a constant yields fitting accuracy that is inferior to the previously proposed logarithmic parallel scaling law.

# F   Training Loss for Pile and Stack-V2-Python

Figure 7 illustrates the curve of loss versus data size in our scaling law experiments. It clearly shows that scaling $P$ yields benefits, regardless of the data scale.

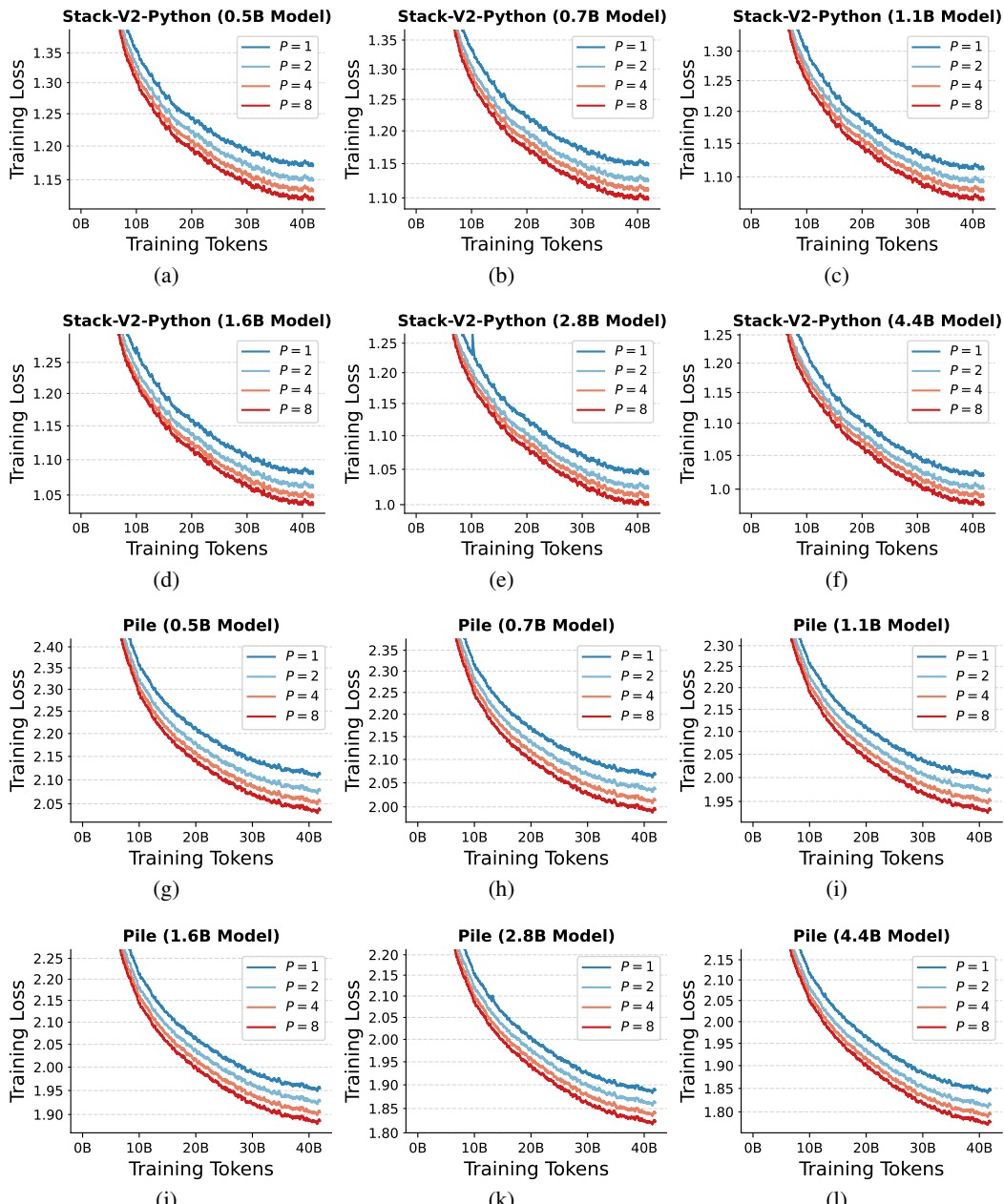

Figure 7: Training loss for the Stack-V2-Python and the Pile, smoothing with 0.98 exponential moving average.

## G Downstream Datasets

Table 15: HumanEval Pass@1 (%)

| $N$ | 0.5B | 0.7B | 1.1B | 1.6B | 2.8B | 4.4B |
|-----|------|------|------|------|------|------|
| $P = 1$ | 12.8 | 15.9 | 17.7 | 18.3 | 18.3 | 19.5 |
| $P = 2$ | 15.9 | 20.7 | 20.1 | 19.5 | 18.9 | 24.4 |
| $P = 4$ | 15.2 | 17.1 | 18.3 | 18.3 | 22.0 | 20.7 |
| $P = 8$ | 18.9 | 18.3 | 21.3 | 18.3 | 21.3 | 25.0 |

Table 16: HumanEval+ Pass@1 (%)

| $N$ | 0.5B | 0.7B | 1.1B | 1.6B | 2.8B | 4.4B |
|-----|------|------|------|------|------|------|
| $P = 1$ | 11.0 | 13.4 | 15.9 | 15.9 | 15.9 | 16.5 |
| $P = 2$ | 13.4 | 17.7 | 17.7 | 16.5 | 16.5 | 21.3 |
| $P = 4$ | 14.0 | 14.6 | 15.9 | 15.9 | 18.9 | 18.3 |
| $P = 8$ | 15.9 | 15.2 | 18.3 | 16.5 | 19.5 | 20.7 |

Table 17: MBPP Pass@1 (%)

| $N$ | 0.5B | 0.7B | 1.1B | 1.6B | 2.8B | 4.4B |
|-----|------|------|------|------|------|------|
| $P = 1$ | 27.8 | 30.2 | 31.5 | 36.0 | 40.5 | 45.8 |
| $P = 2$ | 34.4 | 33.6 | 36.8 | 42.6 | 47.4 | 47.1 |
| $P = 4$ | 35.2 | 33.9 | 39.4 | 40.7 | 45.5 | 50.3 |
| $P = 8$ | 33.3 | 36.0 | 39.9 | 45.5 | 47.4 | 48.4 |

Table 18: MBPP+ Pass@1 (%)

| $N$ | 0.5B | 0.7B | 1.1B | 1.6B | 2.8B | 4.4B |
|-----|------|------|------|------|------|------|
| $P = 1$ | 23.5 | 26.2 | 25.7 | 31.2 | 34.7 | 38.4 |
| $P = 2$ | 27.0 | 27.8 | 30.7 | 35.4 | 39.4 | 38.9 |
| $P = 4$ | 28.0 | 27.2 | 32.3 | 32.5 | 37.3 | 40.5 |
| $P = 8$ | 29.1 | 29.1 | 33.6 | 38.9 | 40.5 | 42.3 |

Table 19: HumanEval Pass@10 (%)

| $N$ | 0.5B | 0.7B | 1.1B | 1.6B | 2.8B | 4.4B |
|-----|------|------|------|------|------|------|
| $P = 1$ | 25.0 | 21.3 | 29.9 | 29.9 | 32.9 | 37.2 |
| $P = 2$ | 25.0 | 27.4 | 28.0 | 35.4 | 34.8 | 40.2 |
| $P = 4$ | 26.2 | 30.5 | 28.0 | 36.0 | 38.4 | 40.2 |
| $P = 8$ | 29.9 | 32.3 | 31.7 | 37.2 | 38.4 | 47.0 |

Table 20: HumanEval+ Pass@10 (%)

| $N$ | 0.5B | 0.7B | 1.1B | 1.6B | 2.8B | 4.4B |
|-----|------|------|------|------|------|------|
| $P = 1$ | 23.8 | 21.3 | 26.8 | 26.8 | 29.9 | 33.5 |
| $P = 2$ | 22.0 | 25.6 | 24.4 | 31.7 | 32.3 | 36.0 |
| $P = 4$ | 24.4 | 27.4 | 25.6 | 33.5 | 33.5 | 34.8 |
| $P = 8$ | 27.4 | 29.9 | 29.9 | 32.9 | 36.0 | 42.7 |

Table 21: MBPP Pass@10 (%)

| $N$ | 0.5B | 0.7B | 1.1B | 1.6B | 2.8B | 4.4B |
|-----|------|------|------|------|------|------|
| $P = 1$ | 49.2 | 54.0 | 57.7 | 61.4 | 68.3 | 66.4 |
| $P = 2$ | 57.9 | 57.7 | 60.3 | 64.0 | 67.7 | 72.0 |
| $P = 4$ | 54.0 | 59.8 | 61.1 | 66.9 | 70.9 | 73.5 |
| $P = 8$ | 57.7 | 60.3 | 66.1 | 67.5 | 72.8 | 75.1 |

Table 22: MBPP+ Pass@10 (%)

| $N$ | 0.5B | 0.7B | 1.1B | 1.6B | 2.8B | 4.4B |
|-----|------|------|------|------|------|------|
| $P = 1$ | 40.2 | 44.7 | 47.9 | 51.6 | 54.5 | 56.3 |
| $P = 2$ | 46.8 | 48.4 | 50.5 | 54.2 | 58.2 | 60.6 |
| $P = 4$ | 43.7 | 49.5 | 52.1 | 56.6 | 59.0 | 62.7 |
| $P = 8$ | 46.0 | 50.8 | 56.6 | 56.1 | 60.6 | 62.2 |

Table 23: WinoGrande Performance (%)

| $N$ | 0.5B | 0.7B | 1.1B | 1.6B | 2.8B | 4.4B |
|-----|------|------|------|------|------|------|
| $P = 1$ | 51.9 | 51.2 | 51.6 | 52.2 | 53.5 | 54.7 |
| $P = 2$ | 51.0 | 51.4 | 53.0 | 53.3 | 56.0 | 57.4 |
| $P = 4$ | 52.4 | 53.0 | 53.5 | 54.5 | 56.7 | 56.4 |
| $P = 8$ | 51.7 | 53.6 | 55.0 | 53.4 | 55.6 | 56.9 |

Table 24: Hellaswag Performance (%)

| $N$ | 0.5B | 0.7B | 1.1B | 1.6B | 2.8B | 4.4B |
|-----|------|------|------|------|------|------|
| $P = 1$ | 35.7 | 37.4 | 40.1 | 42.6 | 46.7 | 49.3 |
| $P = 2$ | 36.8 | 38.4 | 41.3 | 44.5 | 48.4 | 51.9 |
| $P = 4$ | 37.4 | 39.4 | 42.9 | 45.7 | 50.0 | 53.8 |
| $P = 8$ | 38.6 | 40.6 | 44.1 | 46.8 | 51.0 | 54.6 |

Table 25: OpenBookQA Performance (%)

| $N$ | 0.5B | 0.7B | 1.1B | 1.6B | 2.8B | 4.4B |
|-----|------|------|------|------|------|------|
| $P = 1$ | 26.0 | 28.8 | 28.2 | 28.0 | 29.0 | 32.4 |
| $P = 2$ | 26.8 | 26.6 | 27.8 | 29.8 | 30.6 | 29.8 |
| $P = 4$ | 26.6 | 29.0 | 29.8 | 29.4 | 31.0 | 32.0 |
| $P = 8$ | 26.8 | 27.2 | 29.4 | 31.0 | 31.6 | 30.6 |

Table 26: PiQA Performance (%)

| $N$ | 0.5B | 0.7B | 1.1B | 1.6B | 2.8B | 4.4B |
|-----|------|------|------|------|------|------|
| $P = 1$ | 65.0 | 65.8 | 66.9 | 67.5 | 68.8 | 69.5 |
| $P = 2$ | 65.7 | 66.8 | 67.4 | 68.5 | 70.5 | 71.3 |
| $P = 4$ | 65.8 | 66.7 | 68.0 | 68.8 | 70.6 | 72.1 |
| $P = 8$ | 66.5 | 66.9 | 67.8 | 69.5 | 70.9 | 71.5 |

**Setup for 42B token experiments** For HumanEval(+) [12] and MBPP(+) [3], we use the EvalPlus framework [53] for evaluation, where Pass@1 employs greedy decoding and Pass@10 employs a temperature of 0.8. Considering that the pretrained base model cannot follow instructions, we use the direct completion format. For general tasks, we employ lm-eval harness [6] and report normalized accuracy when provided. The number of few-shot mostly follows existing research configurations. Benchmarks include: WinoGrande (5-shot, Sakaguchi et al. [74]), Hellaswag (10-shot, Zellers et al. [98]), OpenBookQA (5-shot, Mihaylov et al. [62]), PiQA (5-shot, Bisk et al. [7]), ARC-Easy and ARC-Challenge (25-shot, Clark et al. [15]; we reporting the average of both), and SciQ (3-shot, Welbl et al. [92]).

| Table 27: ARC Performance (%) | | | | | | |
|---|---|---|---|---|---|---|
| $N$ | 0.5B | 0.7B | 1.1B | 1.6B | 2.8B | 4.4B |
| $P = 1$ | 36.9 | 38.5 | 40.5 | 42.1 | 44.7 | 46.4 |
| $P = 2$ | 38.6 | 39.9 | 40.9 | 43.2 | 45.9 | 49.1 |
| $P = 4$ | 39.1 | 39.9 | 41.1 | 44.2 | 47.4 | 48.8 |
| $P = 8$ | 39.4 | 39.8 | 42.0 | 44.8 | 48.2 | 49.4 |

| Table 28: SciQ Performance (%) | | | | | | |
|---|---|---|---|---|---|---|
| $N$ | 0.5B | 0.7B | 1.1B | 1.6B | 2.8B | 4.4B |
| $P = 1$ | 78.9 | 81.7 | 85.2 | 86.1 | 88.5 | 91.0 |
| $P = 2$ | 80.8 | 83.2 | 84.0 | 87.3 | 90.5 | 91.4 |
| $P = 4$ | 82.3 | 82.7 | 84.4 | 87.2 | 91.2 | 91.7 |
| $P = 8$ | 81.3 | 83.0 | 86.9 | 88.9 | 91.2 | 94.2 |

**Setup for 1T token experiments** In the 1T token experiments, we use more challenging datasets for comprehensive testing, including MMLU (5-shot, Hendrycks et al. [30]) and RACE (4-shot, Lai et al. [43]). In addition, we introduce mathematics reasoning datasets, including GSM8K (4-shot, Cobbe et al. [17]), GSM8K-CoT (8-shot), and MATH (4-shot, Hendrycks et al. [31]; using the Minerva evaluation rules [47]). When evaluating the Instruct model, we also use IFEval (0-shot, Zhou et al. [103]) and report the average among four metrics provided by lm-eval harness.

Tables 15 to 28 show the downstream task performance in the 42B token experiments. We report the average of these performances in the main text (Tables 2 and 3).

## H    Compared with Inference-time Parallel Scaling

Some training-free inference-time scaling methods also expand parallel computation during the inference phase, with Beam Search and Majority Voting being representative methods. We applied Beam Search and Majority Voting to the trained Baseline-1.8B and compared it with PARSCALE to emphasize the importance of expanding parallel computation during the training phase. It is worth mentioning that these methods cannot be applied to the general tasks in Table 4 because these tasks primarily evaluate next-token prediction.

Table 29: Comparison with Beam-Search.

| Number of Parallels | Method | GSM8K | GSM8K-CoT | Minerva MATH |
|---|---|---|---|---|
| 1 | - | 28.7 | 35.9 | 12.0 |
| 2 | PARSCALE | 32.6 | 35.6 | 13.0 |
| 2 | Beam-Search | 29.3 | 37.8 | 13.6 |
| 2 | Majority Voting (temp=0.2) | 28.2 | - | 12.0 |
| 2 | Majority Voting (temp=0.8) | 19.7 | - | 7.5 |
| 4 | PARSCALE | 34.7 | 40.8 | 14.5 |
| 4 | Beam-Search | 27.7 | 37.8 | 12.5 |
| 4 | Majority Voting (temp=0.2) | 31.5 | - | 12.6 |
| 4 | Majority Voting (temp=0.8) | 25.7 | - | 8.5 |
| 8 | PARSCALE | 38.4 | 43.7 | 16.4 |
| 8 | Beam-Search | 22.5 | 30.1 | 10.0 |
| 8 | Majority Voting (temp=0.2) | 32.8 | - | 14.5 |
| 8 | Majority Voting (temp=0.8) | 32.0 | - | 10.8 |

The results are shown in Table 29. It can be observed that Beam Search is only effective when $n\_beams = 2$. As $n\_beams$ increases, the performance of Beam Search actually decreases. This indicates that, without a strong verifier, it is difficult for LLM alone to select the correct result from multiple sampling results. This aligns with the finding in [81]. This experiment further emphasizes the importance of expanding parallel computation in both the training stage and inference stage.

## I    Instruction Tuning

We follow standard practice to post-train the base models, to explore whether PARSCALE can enhance performance during the post-training stage. We conducted instruction tuning on four checkpoints ($P \in \{1, 2, 4, 8\}$) from the previous pre-training step, increasing the sequence length from 2048 to 8192 and the RoPE base from 10,000 to 100,000, while keeping other hyperparameters constant. We

Table 30: Comparison of the performance of different Instruct models, where the few-shot examples are treated as a multi-turn conversation.

|  | IFEval 0-shot | MMLU 5-shot | GSM8K 4-shot |
|---|---|---|---|
| SmolLM-1.7B-Inst | 16.3 | 28.4 | 2.0 |
| Baseline-Inst ($P = 1$) | 54.1 | 34.2 | 50.3 |
| PARSCALE-Inst ($P = 2$) | 55.8 | 35.1 | 55.3 |
| PARSCALE-Inst ($P = 4$) | 58.4 | 38.2 | 54.8 |
| PARSCALE-Inst ($P = 8$) | **59.5** | **41.7** | **56.1** |

used 1 million SmolTalk [2] as the instruction tuning data and trained for 2 epochs. We refer to these instruction models as **PARSCALE-Inst**.

The experimental results in Table 30 show that when $P$ increases from 1 to 8, our method achieves a 5% improvement on the instruction-following benchmark IFEval, along with substantial gains in the general task MMLU and the reasoning task GSM8K. This demonstrates that the proposed PARSCALE performs effectively during the post-training phase.

## J    Continual Pre-training Qwen-2.5 3B Model

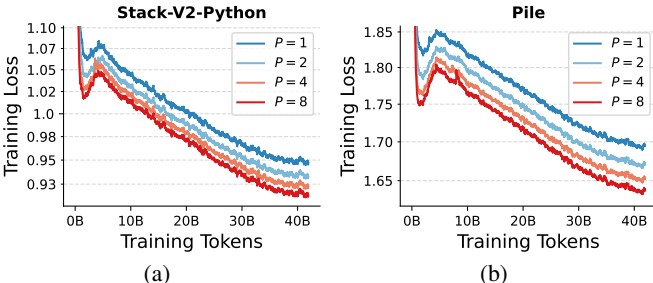

Figure 8: Loss for continual pre-training the Qwen-2.5-3B model on the two datasets.

Figures 8(a) and 8(b) illustrates the training loss after continuous pre-training on Stack-V2 (Python) and Pile. Notably, Qwen2.5 has already been pre-trained on 18T of data, which possibly have significant overlap with both Pile and Stack-V2. This demonstrates that improvements can still be achieved even with a thoroughly trained foundation model and commonly used training datasets.

## K    Discussion and Future Work

**More Interpretation of Scaling Law Claims**   Our analysis relies on fitting a log power-law relationship to empirical scaling data using Huber loss, which is less sensitive to outliers than standard squared loss. While this approach improves robustness, it does not, by itself, constitute rigorous statistical evidence for the existence of a true power-law distribution in the underlying data. As emphasized by Clauset et al. [16], identifying power laws in empirical data is notoriously challenging and requires careful model comparison, goodness-of-fit testing, and consideration of alternative functional forms (e.g., log-normal or exponential). Future work could strengthen these conclusions by applying more rigorous scaling law detection protocols as outlined in the statistical literature.

**Training Inference-Optimal Language Models**   Chinchilla [36] explored the scaling law to determine the training-optimal amounts for parameters and training data under a training FLOP budget. On the other hands, modern LLMs are increasingly interested on inference-optimal models. Some practitioners use much more data than the Chinchilla recommendation to train small models

due to their high inference efficiency [70, 2, 76]. Recent inference-time scaling efforts attempt to provide a computation-optimal strategy during the inference phase [95, 80], but most rely on specific scenarios and datasets. Leveraging the proposed PARSCALE, determining how to allocate the number of parameters and parallel computation under various inference budgets (e.g., memory, latency, and batch size) to extend inference-optimal scaling laws [76] is a promising direction.

**Further Theoretical Analysis for Parallel Scaling Laws**   One of our contributions is quantitatively computing the impact of parameters and computation on model capability. Although we present some theoretical results (Proposition 1), the challenge of directly modeling DIVERSITY limits us to using extensive experiments to fit parallel scaling laws. Why the diversity is related to $\log P$, is there a growth rate that exceeds $\mathcal{O}(\log P)$, and whether there is a performance upper bound for $P \gg 8$, remain open questions.

**Optimal Division Point of Two-Stage Strategy**   Considering that PARSCALE is inefficient in the training phase, we introduced a two-stage strategy and found that LLMs can still learn to leverage parallel computation for better capacity with relatively few tokens. We currently employ a 1T vs. 20B tokens as the division point. Whether there is a more optimal division strategy and its trade-off with performance is also an interesting research direction.

**Application to MoE Architecture**   Similar to the method proposed by Geiping et al. [27], PARSCALE is a computation-heavy (but more efficient) strategy. This is somewhat complementary to sparse MoE [23, 77], which is parameter-heavy. Considering that MoE is latency-friendly while PARSCALE is memory-friendly, exploring whether their combination can yield more efficient and high-performing models is worth investigating.

**Application to Other Machine Learning Domains**   Although we focus on language models, PARSCALE is a more general method that can be applied to any model architecture, training algorithm, and training data. Exploring PARSCALE in other areas and even proposing new scaling laws is also a promising direction for the future.

## L    Broader Impacts

LLMs present certain potential risks, such as generating offensive language, perpetuating social biases, and leaking private information. While the public release of all our models and the insights we provide to scaling LLMs could inadvertently contribute to the proliferation of these harms, it is important to note that there are already larger and more capable models freely available that can also be used in harmful ways. We believe that the benefits of open-source releasing our models and research can outweigh the associated risks.

## M    Visualization for Different Parallel Streams

In the output aggregation, we assign different dynamic weights to different streams. These weights indicate the contribution ratio of different parallel streams to the next token prediction. We visualize the stream that contributes the most to each token (i.e., the stream with the highest aggregated weight), based on the 4.4B model pre-trained on the Pile. Tokens of the same color indicating that these tokens are primarily contributed by the same stream. Tables 31 to 33 visualize the results. An interesting observation is the *locality* of the colors: tokens in close proximity are often primarily contributed by the same stream, especially when $P$ is relatively small (e.g., $P = 2$).

Table 31: Parallel stream visualization, where paragraphs are sampled from wikitext [61]. Tokens with the same color indicate they are primarily contributed by the same stream.

| | |
|---|---|
| $P = 2$ | Somerset progressed to the second round of the competition after Trinidad and Tobago beat Deccan in the final group match , but lost Trescothick , who flew home after a recurrence of his illness . Wes Durston , who replaced Trescothick in the side , top - scored for Somerset in their next match , making 57 . Only two other players reached double - figures for the county , and the Diamond Eagles chased down the total with eight balls to spare . Somerset went into their final match , against the New South Wales Blues with a slim mathematical chance of progressing , but a strong bowling display from Brett Lee and Stuart Clark restricted Somerset to 111 , which the Australian side reached with ease . |
| $P = 4$ | Somerset progressed to the second round of the competition after Trinidad and Tobago beat Deccan in the final group match , but lost Trescothick , who flew home after a recurrence of his illness . Wes Durston , who replaced Trescothick in the side , top - scored for Somerset in their next match , making 57 . Only two other players reached double - figures for the county , and the Diamond Eagles chased down the total with eight balls to spare . Somerset went into their final match , against the New South Wales Blues with a slim mathematical chance of progressing , but a strong bowling display from Brett Lee and Stuart Clark restricted Somerset to 111 , which the Australian side reached with ease . |
| $P = 8$ | Somerset progressed to the second round of the competition after Trinidad and Tobago beat Deccan in the final group match , but lost Trescothick , who flew home after a recurrence of his illness . Wes Durston , who replaced Trescothick in the side , top - scored for Somerset in their next match , making 57 . Only two other players reached double - figures for the county , and the Diamond Eagles chased down the total with eight balls to spare . Somerset went into their final match , against the New South Wales Blues with a slim mathematical chance of progressing , but a strong bowling display from Brett Lee and Stuart Clark restricted Somerset to 111 , which the Australian side reached with ease . |

Table 32: Parallel stream visualization, where paragraphs are sampled from GSM8K [17]. Tokens with the same color indicate they are primarily contributed by the same stream.

| | |
|---|---|
| $P = 2$ | Question: A baker is making bread according to a recipe that requires him to use 3 eggs for every 2 cups of flour. If the baker wants to use up the 6 cups of flour he has remaining in his pantry, how many eggs will he need to use?. Answer: If he uses 6 cups of flour, the baker will be making 6/2 = «6/23» times the normal amount that the recipe describes. Thus, he must use 3*3 = «3*=99 eggs. #### 9 |
| $P = 4$ | Question: A baker is making bread according to a recipe that requires him to use 3 eggs for every 2 cups of flour. If the baker wants to use up the 6 cups of flour he has remaining in his pantry, how many eggs will he need to use?. Answer: If he uses 6 cups of flour, the baker will be making 6/2 = «6/2=» times the normal amount that the recipe describes. Thus, he must use 3*3 = «3*3=9 eggs. #### 9 |
| $P = 8$ | Question: A baker is making bread according to a recipe that requires him to use 3 eggs for every 2 cups of flour. If the baker wants to use up the 6 cups of flour he has remaining in his pantry, how many eggs will he need to use?. Answer: If he uses 6 cups of flour, the baker will be making 6/2 = 6/2=3» times the normal amount that the recipe describes. Thus, he must use 3*3 = «3*3=99 eggs. #### 9 |

Table 33: Parallel stream visualization, where paragraphs are sampled from RACE [43]. Tokens with the same color indicate they are primarily contributed by the same stream.

| | |
|---|---|
| $P = 2$ | The Mysterious Universe By Ellen Jackson and Nic Bishop How did the universe begin? How big is it? What is dark matter? Cosmologist and expert supernova hunter Alex Filippenko hopes that supernovas can help us answer some of these questions. But first we've got to find them! Join Alex and his team as they go on the hunt with huge telescopes and banks of computers. The Time and Space of Uncle Albert By Russell Stannard What would you say if your uncle asked you whether you would like to go into space? You'd say, "When do I leave?", just like the girl in this story. Gedanken is speeding across the universe trying to help her uncle answer some questions, such as "How big is space?" and "Where does gravity come from?" Along the way she also discovers how to get heavier without getting fat, how to live forever without knowing it, and the strange things that can happen when you go really fast. George's Secret Key to the Universe By Lucy Hawking and Stephen Hawking When George chases his pet pig through a hole in the fence, little does he expect that he will soon be riding a comet around Saturn . But just as he discovers the joys of space exploration with the computer Cosmos, which can open doors anywhere in the universe, everything starts to go wrong. |
| $P = 4$ | The Mysterious Universe By Ellen Jackson and Nic Bishop How did the universe begin? How big is it? What is dark matter? Cosmologist and expert supernova hunter Alex Filippenko hopes that supernovas can help us answer some of these questions. But first we've got to find them! Join Alex and his team as they go on the hunt with huge telescopes and banks of computers. The Time and Space of Uncle Albert By Russell Stannard What would you say if your uncle asked you whether you would like to go into space? You'd say, "When do I leave?", just like the girl in this story. Gedanken is speeding across the universe trying to help her uncle answer some questions, such as "How big is space?" and "Where does gravity come from?" Along the way she also discovers how to get heavier without getting fat, how to live forever without knowing it, and the strange things that can happen when you go really fast. George's Secret Key to the Universe By Lucy Hawking and Stephen Hawking When George chases his pet pig through a hole in the fence, little does he expect that he will soon be riding a comet around Saturn . But just as he discovers the joys of space exploration with the computer Cosmos, which can open doors anywhere in the universe, everything starts to go wrong. |
| $P = 8$ | The Mysterious Universe By Ellen Jackson and Nic Bishop How did the universe begin? How big is it? What is dark matter? Cosmologist and expert supernova hunter Alex Filippenko hopes that supernovas can help us answer some of these questions. But first we've got to find them! Join Alex and his team as they go on the hunt with huge telescopes and banks of computers. The Time and Space of Uncle Albert By Russell Stannard What would you say if your uncle asked you whether you would like to go into space? You'd say, "When do I leave?", just like the girl in this story. Gedanken is speeding across the universe trying to help her uncle answer some questions, such as "How big is space?" and "Where does gravity come from?" Along the way she also discovers how to get heavier without getting fat, how to live forever without knowing it, and the strange things that can happen when you go really fast. George's Secret Key to the Universe By Lucy Hawking and Stephen Hawking When George chases his pet pig through a hole in the fence, little does he expect that he will soon be riding a comet around Saturn . But just as he discovers the joys of space exploration with the computer Cosmos, which can open doors anywhere in the universe, everything starts to go wrong. |

