# OpenReview forum: "Parallel Scaling Law for Language Models"
_NeurIPS.cc/2025/Conference — NeurIPS 2025 poster_

### Official Review · Reviewer_T6Gx · 2025-06-23

**Clarity:** 4
**Significance:** 4
**Originality:** 4
**Rating:** 5
**Confidence:** 5

**Summary:**

This paper introduces Parallel Scaling (PARSCALE), a novel scaling strategy for language models that aims to increase parallel computation during both training and inference without significantly increasing the number of model parameters. The core idea is to execute multiple parallel streams of computation with dynamic aggregation of their outputs. The authors propose a theoretical scaling law and demonstrate through large-scale pretraining that PARSCALE achieves performance comparable to parameter scaling while offering substantial reductions in memory usage and latency. This makes the method particularly appealing in resource-constrained environments.

**Questions:**

1. How does PARSCALE’s scalability and efficiency vary with the number of GPUs in a cluster?
2. Has PARSCALE been evaluated on more complex models like GPT or LLaMA, and is it compatible with MoE (Mixture-of-Experts) architectures?
3. Does PARSCALE introduce additional communication overhead in distributed training or inference, and how does this overhead vary with different parallelization strategies (DP, TP, PP) ?

**Ethical Concerns:**

["NO or VERY MINOR ethics concerns only"]

**Final Justification:**

This is a very solid paper. The rebuttal has removed all my concerns. I agree to accept this paper.

**Limitations:**

1. Lacking analysis of how PARSCALE performs across different GPU setups or hardware configurations.
2. Lacking analysis of PARSCALE's communication volume and its scalability across different cluster scales or parallelisms.

**Paper Formatting Concerns:**

No formatting concern.

**Quality:**

3

**Strengths And Weaknesses:**

### Strengths
1. Innovative Scaling Strategy: PARSCALE presents a fresh approach to model scaling by increasing computational parallelism instead of parameter count.
2. Solid Theoretical Foundation: The proposed scaling law is well-formulated and empirically validated, providing strong theoretical support for the method.
3. Efficiency Gains: PARSCALE demonstrates significant improvements in memory and latency efficiency, which is particularly beneficial for deployment on edge devices.
4. Cost-Effective Training: The two-stage training strategy effectively reduces training costs, making large-scale model training more accessible.
5. Strong Empirical Results: The experimental results are compelling, especially for reasoning-heavy tasks such as code generation and mathematical problem solving.
5. Strong Experimental Results: The paper presents compelling experimental evidence, showing significant improvements in performance for reasoning-intensive tasks, particularly in code generation and math.

### Weaknesses
1. Scalability on Larger Models: The paper lacks an in-depth evaluation of PARSCALE’s scalability on very large models and across diverse architectures such as GPT and LLaMA.
2. Hardware Variability: While memory and latency improvements are reported, the paper does not discuss how these benefits hold across different hardware environments (e.g., GPU types or configurations).
3. Multi-GPU Scaling: There is no exploration of how PARSCALE performs with varying numbers of GPUs, which is crucial for understanding its hardware scalability.

---

> ### Author Rebuttal · Authors · 2025-07-31
>
> We would like to extend our sincere gratitude for your thoughtful and constructive feedback on our manuscript. We are pleased that you recognized the core contributions of our paper. We appreciate your acknowledgment of the `Solid Theoretical Foundation` of our proposed scaling law, which is both well-formulated and empirically validated. We are also grateful for your positive feedback on the significant `Efficiency Gains` in memory and latency that ParScale provides, its potential for enabling `Cost-Effective Training`, and the `Strong Empirical Results` we demonstrated.
>
> Below are our responses to your concerns:
>
> > W1: Scalability on Larger Models: The paper lacks an in-depth evaluation of PARSCALE’s scalability on very large models and across diverse architectures such as GPT and LLaMA.
>
> > Q2: Has PARSCALE been evaluated on more complex models like GPT or LLaMA, and is it compatible with MoE (Mixture-of-Experts) architectures?
>
> Thank you for your valuable comments.
> - Larger Model: Our study on models from 0.6B to 4.4B aligns with standard practices in scaling law research. Given the substantial computational cost (over 10^6 GPU hours), we defer training even larger models to future work, which will be guided by our proposed scaling law. As stated in Line 196, we believe that larger models will benefit even more from ParScale.
> - Architectures (GPT/Llama): We chose Qwen as it is a representative modern Transformer, sharing its core design with GPT and Llama. This architecture is not less complex than GPT and Llama. Our conclusions are thus expected to be broadly applicable across these standard architectures.
> - MoE Compatibility: ParScale is fully compatible with MoE and we plan to investigate their combination, as we stated in the future work section (Line 1014).
>
> > W2: Hardware Variability: While memory and latency improvements are reported, the paper does not discuss how these benefits hold across different hardware environments (e.g., GPU types or configurations).
>
> - Thank you for this important point on hardware variability. The memory and latency benefits of ParScale stem from fundamental principles—improving arithmetic intensity (the ratio of compute to memory access) and reducing the overall memory footprint. **Since these principles are universal to all modern parallel processors, the advantages are not tied to a specific GPU architecture.** While our experiments were conducted on industry-standard NVIDIA A100 GPUs, the observed efficiency gains are expected to be robust and generalizable across different hardware environments, including various GPU types and configurations.
>
> > W3: Multi-GPU Scaling: There is no exploration of how PARSCALE performs with varying numbers of GPUs, which is crucial for understanding its hardware scalability.
>
> > Q1: How does PARSCALE’s scalability and efficiency vary with the number of GPUs in a cluster?
>
> > Q3: Does PARSCALE introduce additional communication overhead in distributed training or inference, and how does this overhead vary with different parallelization strategies (DP, TP, PP) ?
>
> - Thank you for these insightful questions regarding hardware scalability. ParScale is specifically designed to be highly efficient in multi-GPU settings because it operates on the Data Parallelism (DP) dimension. It works by partitioning a single data sample into independent parallel streams that are processed without any cross-communication (except for the input transformation and output aggregation).
> - Consequently, unlike TP or PP which introduce significant communication overhead, ParScale adds nearly no additional communication cost. Its scalability profile is therefore similar to that of standard DP, which is well-known for its high efficiency and near-linear scaling with the number of GPUs.
>
> ---
>
> Thank you again for your encouraging feedback. We hope our response could fully address your concerns.

---

> > ### Comment · Reviewer_T6Gx · 2025-08-05
> >
> > Thank you for responding to my questions. I do not have additional questions or concerns. I believe this is a very solid paper and choose to keep my initial recommendation unchanged.

---

### Official Review · Reviewer_UM4q · 2025-06-30

**Clarity:** 4
**Significance:** 3
**Originality:** 3
**Rating:** 5
**Confidence:** 3

**Summary:**

This paper presents a brand-new way of thinking about Scaling Laws, focusing more on expanding a model's capabilities (as seen in the loss curve) by increasing the amount of computation rather than the computational space (i.e., the number of model parameters).

**Questions:**

1. Under current application conditions, can Parallel Scaling demonstrate a sufficient performance advantage over MoE and Inference-Time Scaling? If not now, is there hope for it in the future?

2.  Under current conditions, is the Parallel Scaling method robust enough? Compared to sample-level scaling like self-consistency, and considering the method's huge resource consumption (even just in post-training), is the proposed prefix extension more cost-effective than no-training alternatives (like using Dropout) or using MoE for parallel scaling?

**Ethical Concerns:**

["NO or VERY MINOR ethics concerns only"]

**Final Justification:**

I sincerely thank the authors for their detailed experiments. The provided experimental results have fully addressed all of my concerns. I appreciate that the authors have offered clear, empirical evidence. This is a strong paper, and I recommend it for acceptance.

**Limitations:**

yes.

**Paper Formatting Concerns:**

None.

**Quality:**

2

**Strengths And Weaknesses:**

**Strengths**

This paper dedicates significant resources to discussing the Parallel Scaling Law, which I believe is a very promising approach. Recently, the decoding and training speeds of LLMs are often bottlenecked by communication rather than raw computation speed. This makes scaling up the amount of computation particularly important, especially in a context where GPU computational performance is growing rapidly while communication bandwidth and GPU memory remain significant constraints. I was impressed by the clear and detailed experiments in this paper that introduce this novel scaling method.


**Weaknesses**

The paper lacks a more in-depth comparison with methods like Self-consistency. Although Self-consistency is, to some extent, a form of sample-level scaling, it has shown benefits in many areas, including reasoning and question answering. On the other hand, when considering a universal method like Universal Self-Consistency for Large Language Model Generation, I find it difficult to think of a generative AI scenario where the Parallel Scaling Law could be applied, aside from real-time streaming generation.

The paper explores, at most, a scenario with P=8. As far as I know, methods like Self-consistency often have an upper limit; their performance growth slows down significantly once the number of samples reaches around 1000. I believe it is too early to determine the curve's trend for the Parallel Scaling Law with P=8.

From a cost perspective, the computation in this paper is equivalent to P x L x T, whereas for inference scaling law, it is L x T'. When the computational cost at inference time is comparable—for instance, generating 1000 words with P=8 versus generating 8000 words with P=1 (the latter might have a longer wait time, but I believe this setup is a reasonable approximation when considering resource consumption and throughput in high-concurrency scenarios)—I am curious which would perform better. Considering that in the MATH dataset, the Qwen3-235B-A22B model improved from a base of 71.84 to 98 through inference scaling law, or that the same Qwen3-235B-A22B model, using a "thinking" mode, improved its MATH score from 91.2 (non-thinking) to 98, and its CodeForces score from 1387 to 2056, I think it's difficult for the Parallel Scaling Law, based on Table 4, to demonstrate this level of performance improvement.

I am not sure if the scaling method proposed in this paper is effective enough, and the paper itself lacks corresponding comparative experiments. For example, if I use a baseline model with P=1, I could use a Dropout-like approach where I input the same P=8 input texts during each forward pass and then, in the final layer of the decoding process, merge these eight parallel Hidden States (e.g., by averaging them) to predict the next new token. This method requires no training, but I believe it could still yield similar results.

As another example, one could also test this by directly evaluating a Mixture of Experts (MoE) model. For instance, the DeepSeek-V3/R1 model activates 8 experts per token. Therefore, one could sequentially activate P=1, P=2, P=4, P=8, P=16, P=32 experts, which would also allow for scaling without the need for training.

---

> ### Author Rebuttal · Authors · 2025-07-31
>
> We would like to express our sincere gratitude for your thorough and insightful review of our manuscript. We are particularly encouraged that you found our work on the Parallel Scaling Law to be a promising approach for `addressing the communication bottlenecks` in modern LLMs. We appreciate you recognizing the value of our `clear and detailed experiments` and acknowledging our contribution in introducing this `novel scaling method`. Below are our responses:
>
> > W1: The paper lacks a more in-depth comparison with methods like Self-consistency. Although Self-consistency is, to some extent, a form of sample-level scaling, it has shown benefits in many areas, including reasoning and question answering. On the other hand, when considering a universal method like Universal Self-Consistency for Large Language Model Generation, I find it difficult to think of a generative AI scenario where the Parallel Scaling Law could be applied, aside from real-time streaming generation.
>
> - Thank you for your suggestion. The table below presents the performance comparison on GSM8K and MATH using ParScale-1.8B (P=1) with self-consistency (SC), showing that ParScale consistantly performs better than SC.
>
> |  | ParScale | SC (temperature = 0.2) | SC (temperature = 0.8) |
> | --- | --- | --- | --- |
> | **GSM8K** |  |  |  |
> | P=2 | **32.6** | 28.2 | 19.7 |
> | P=4 | **34.7** | 31.5 | 25.7 |
> | P=8 | **38.4** | 32.8 | 32.0 |
> | **MATH** |  |  |  |
> | P=2 | **13.0** | 12.0 | 7.5 |
> | P=4 | **14.5** | 12.6 | 8.5 |
> | P=8 | **16.4** | 14.5 | 10.8 |
>
> We would also like to provide a more in-depth comparison of these two methods:
>
> - **Foundational Method vs. Inference Technique**: ParScale is a foundational method that enhances a model's intrinsic capacity, making it directly comparable to parameter scaling. In contrast, SC is an inference-time sampling technique applied as a post-processing step. They operate at different levels.
> - **Broader and Distinct Applicability**. We respectfully clarify that ParScale can be used for critical generative AI scenarios where methods like SC are impractical, such as real-time streaming generation and open-ended dialog. Additionally, unlike SC, ParScale improves the single-shot generation quality. This is crucial for tasks requiring a single, high-quality output, or in specific architectural configurations (e.g., when P=2) where a simple majority vote is not applicable.
> - **Generalizable Principle**. The concept of parallel scaling is a general architectural principle with the potential to extend beyond LLMs to other domains like diffusion models and MLPs, particularly in memory-limited contexts. SC is limited for text generation.
>
> > W2: The paper explores, at most, a scenario with P=8. As far as I know, methods like Self-consistency often have an upper limit; their performance growth slows down significantly once the number of samples reaches around 1000. I believe it is too early to determine the curve's trend for the Parallel Scaling Law with P=8.
>
> - Thank you for your insightful comment. We agree that the performance for P >> 8 is an interesting open question and we consider it a direction for future work due to our computation budget (our experiments already take up above 10^6 GPU-h). We have mentioned this in line 1007 (Future work).
>
> > W3: From a cost perspective, the computation in this paper is equivalent to P x L x T, whereas for inference scaling law, it is L x T'. When the computational cost at inference time is comparable—for instance, generating 1000 words with P=8 versus generating 8000 words with P=1 (the latter might have a longer wait time, but I believe this setup is a reasonable approximation when considering resource consumption and throughput in high-concurrency scenarios)—I am curious which would perform better. Considering that in the MATH dataset, the Qwen3-235B-A22B model improved from a base of 71.84 to 98 through inference scaling law, or that the same Qwen3-235B-A22B model, using a "thinking" mode, improved its MATH score from 91.2 (non-thinking) to 98, and its CodeForces score from 1387 to 2056, I think it's difficult for the Parallel Scaling Law, based on Table 4, to demonstrate this level of performance improvement.
>
> - Thanks for you valuable and interesting analysis! First, we kindly argue that **direct controlling cost for comparison is challenging.** The computational cost of inference time scaling is unpredictable due to the variable token count (T'), whereas our method's cost is deterministic and controllable via the number of parallel instances (P).
> - Second, we wish to emphasize that parallel scaling and serial scaling (e.g., extending reasoning tokens) are **not mutually exclusive**. As shown in our Table 4, combining CoT with ParScale yields further improvements on GSM8K. For future work, we plan to investigate the integration of ParScale with reasoning models like Qwen3.
> - Furthermore, we would like to clarify the concept of "cost." **While FLOPs is a key metric, it does not capture the full picture of practical resource utilization.** ParScale fully leverages hardware parallelism, which serial generation inherently underutilizes. This provides a crucial efficiency advantage and this benefit will grow with more powerful parallelization hardware.
> - Finally, from a generality perspective, our method is a universal technique that applies to any model structure, unlike the inference-time scaling which is only applicable to text generation. This offers broader applicability than approaches relying on specialized training data or architecturally-ingrained thinking modes.
>
> > W4: I am not sure if the scaling method proposed in this paper is effective enough, and the paper itself lacks corresponding comparative experiments. For example, if I use a baseline model with P=1, I could use a Dropout-like approach where I input the same P=8 input texts during each forward pass and then, in the final layer of the decoding process, merge these eight parallel Hidden States (e.g., by averaging them) to predict the next new token. This method requires no training, but I believe it could still yield similar results.
>
> - Thank you for your suggestion. Regarding the mentioned baseline (i.e., MC Dropout), **it is shown in the literature [1] that it yields only marginal improvements and is significantly underperformed by training-based methods.** To further address your concern, we trained a 0.6B model on Stack V2 using MC Dropout and recorded its loss, as shown in the table below:
>
> | P | ParScale (no dropout) | MC Dropout (dropout=0.1) |
> | --- | --- | --- |
> | 1 | 1.1518 | 1.2064 |
> | 2 | 1.1276 | 1.2016 |
> | 4 | 1.1145 | 1.1923 |
> | 8 | 1.1019 | 1.1909 |
>
> The results above once again demonstrate the importance of scaling parallel computation during the training phase. It is noteworthy that the pre-training recipes for most modern LLMs (like Qwen and Llama) have omitted dropout, making it difficult to apply MC Dropout.
>
>
> > W5: As another example, one could also test this by directly evaluating a Mixture of Experts (MoE) model. For instance, the DeepSeek-V3/R1 model activates 8 experts per token. Therefore, one could sequentially activate P=1, P=2, P=4, P=8, P=16, P=32 experts, which would also allow for scaling without the need for training.
> - Thank you for your suggestion. We conducted this experiment on Qwen3 30B-A2B MoE model, which has 128 experts and similarly activates 8 experts per token. The table below presents its performance on GSM8K, MATH, and EvalPlus:
>
> | Activated Experts | GSM8K | MATH | EvalPlus |
> | --- | --- | --- | --- |
> | 8 (original) | 91.2 | 32.7 | 71.3 |
> | 16 | 91.2 | 29.4 | 71.1 |
> | 32 | 84.9 | 28.7 | 66.1 |
>
> As can be seen, performance actually degrades as the number of activated experts increases. This demonstrates that **the best results are achieved only when the training and inference setups are kept consistent.**
>
> > Q1: Under current application conditions, can Parallel Scaling demonstrate a sufficient performance advantage over MoE and Inference-Time Scaling? If not now, is there hope for it in the future?
>
> - As stated in the response to W3, Parallel Scaling and Parameter/Inference-Time Scaling are orthogonal and can be combined. In our paper, we demonstrate that Parallel Scaling offers a greater inference cost advantage, particularly when the batch size is small. We also hope that future work could further raise the value of $k$ (i.e., the performance gain for ParScale) and even break the bound of $O(\log P)$.
>
> > Q2: Under current conditions, is the Parallel Scaling method robust enough? Compared to sample-level scaling like self-consistency, and considering the method's huge resource consumption (even just in post-training), is the proposed prefix extension more cost-effective than no-training alternatives (like using Dropout) or using MoE for parallel scaling?
>
> - We have used extensive experiments (including Scaling Laws and downstream performance) to demonstrate the robustness of Parallel Scaling. Regarding the other baselines you mentioned, we have already established the importance of scaling parallel computation during the training phase in our response to W4 and W5.
>
> ---
>
> Thank you again for your invaluable feedback. We hope our response could effectively address your concerns. Given the extend of our experimental effort, we would be grateful if you would reconsider your evaluation of our manuscript.
>
> [1] LoRA Emsembles for Large Lanuage Model Fine-Tuning.

---

> > ### Comment · Reviewer_UM4q · 2025-07-31
> >
> > I sincerely thank the authors for their detailed experiments. The provided experimental results have fully addressed all of my concerns. I appreciate that the authors have offered clear, empirical evidence. This is a strong paper, and I recommend it for acceptance.

---

> > > ### Comment · Area_Chair_TWv6 · 2025-08-05
> > > **Discussion**
> > >
> > > Dear Reviewer UM4q,
> > >
> > > The authors have responded to your concerns. How does their response change your view of the paper? If it does not, please clarify what the authors can do to address your concerns. If it does, please consider adjusting your score based on their response.
> > >
> > > Your AC

---

### Official Review · Reviewer_8hYJ · 2025-07-01

**Clarity:** 3
**Significance:** 3
**Originality:** 3
**Rating:** 4
**Confidence:** 4

**Summary:**

This paper introduces a novel scaling paradigm for language models called Parallel Scaling (PARSCALE), which serves as an alternative to traditional parameter scaling and inference-time scaling. The core method involves applying P diverse, learnable transformations to an input, processing these P versions in parallel through a single shared-weight model, and then dynamically aggregating the outputs to produce a final result. This approach effectively scales the amount of parallel computation during both training and inference by reusing existing model parameters, thereby avoiding the significant space or time costs associated with conventional scaling methods.

The primary contribution is the proposal and empirical validation of a new "parallel scaling law," which finds that scaling computation by P parallel streams is analogous to scaling the model's parameters by a factor of O(log P). The authors demonstrate that for an equivalent performance gain, PARSCALE is substantially more efficient during inference, showing significant reductions in memory and latency increases compared to parameter scaling. Furthermore, the paper introduces a practical two-stage training strategy to mitigate high training costs and shows that PARSCALE can be effectively applied to off-the-shelf models, enabling dynamic adjustment of model capability at deployment.

**Questions:**

* Clarification on "Similar Effects as Parameter Scaling": The paper claims that PARSCALE has "similar effects as parameter scaling." I would appreciate clarification on this point. Does this mean that for an equivalent increase in total computation (FLOPs), scaling P yields a comparable performance improvement to scaling parameters? Or does it primarily mean that the performance improvements from scaling P can be modeled by a scaling law with a similar mathematical form (i.e., a power law) to that of parameter scaling?
* Optimizing KV Cache Memory Efficiency: The PARSCALE approach can be viewed as increasing the effective batch size of computation over a shared set of parameters, which successfully improves the compute-to-memory-access ratio for the model weights. However, as implemented, the KV cache size scales linearly with P, which could become the next memory bottleneck, especially for long contexts. Have the authors considered or explored strategies to optimize this? For example, could the P streams potentially share or contribute to a common KV cache, particularly for the initial shared context tokens, to further improve memory efficiency?

**Ethical Concerns:**

["NO or VERY MINOR ethics concerns only"]

**Limitations:**

Yes

**Quality:**

3

**Strengths And Weaknesses:**

Strength：

* The paper is supported by a comprehensive suite of experiments. The authors have done a thorough job of validating the proposed parallel scaling law and demonstrating the method's effectiveness across a wide range of model sizes, datasets, and training regimes.
* The proposed method is elegant in its simplicity and directness. By promoting parameter reuse, it effectively improves the compute-to-memory-access ratio, a critical bottleneck for LLM inference. This characteristic makes the approach highly practical and gives it significant potential for deployment in resource-constrained scenarios, such as on-device and edge computing.

Weakness：

* Lack of Comparison with Alternative Parallel Architectures: The paper compellingly demonstrates the benefits of PARSCALE's "end-to-end" parallel design. However, the analysis could be strengthened by comparing it against other potential parallel computation strategies. For instance, what would be the effect of introducing information exchange between the P streams at intermediate layers? A more tightly-coupled design could involve expanding hidden states into P branches within each layer, performing parallel computations, and then fusing them before the next layer. A discussion or empirical comparison—analyzing both model performance and hardware efficiency (e.g., latency under different batching scenarios)—would help clarify whether PARSCALE's approach of P independent forward passes is inherently more advantageous than these more integrated alternatives.
* Absence of a "Brute-Force" Upper Bound: The paper compares PARSCALE to parameter-scaled models with better inference efficiency. However, a crucial baseline is missing: a standard dense model that is P-times wider, which would have a roughly equivalent computational budget (FLOPs) to a PARSCALE model with P streams. While this "wide" model would be highly inefficient due to its low compute-to-memory-access ratio, including it as a theoretical "upper bound" would be highly informative. It would help quantify the performance trade-off inherent in reusing parameters versus simply adding more, providing a clearer picture of how effectively PARSCALE closes the gap with a brute-force scaling approach for a given computational cost.

---

> ### Author Rebuttal · Authors · 2025-07-31
>
> Thank you for your thoughtful and constructive feedback on our manuscript. We are delighted that you found our work to be supported by a `comprehensive suite of experiments` and that you appreciated the `elegant simplicity and directness` of our proposed method. We are particularly encouraged by your recognition of its potential for improving the compute-to-memory-access ratio, which makes it `highly practical for resource-constrained scenarios`. Below are our responses to your concerns:
>
> > W1: Lack of Comparison with Alternative Parallel Architectures: The paper compellingly demonstrates the benefits of PARSCALE's "end-to-end" parallel design. However, the analysis could be strengthened by comparing it against other potential parallel computation strategies. For instance, what would be the effect of introducing information exchange between the P streams at intermediate layers? A more tightly-coupled design could involve expanding hidden states into P branches within each layer, performing parallel computations, and then fusing them before the next layer. A discussion or empirical comparison—analyzing both model performance and hardware efficiency (e.g., latency under different batching scenarios)—would help clarify whether PARSCALE's approach of P independent forward passes is inherently more advantageous than these more integrated alternatives.
>
> - Thank you for this excellent and thought-provoking suggestion. We would like to first clarify that our primary goal is to demonstrate that scaling parallel computation is a new, effective dimension for improving a model's intrinsic capability. Given our computational budget constraints, we chose the most direct implementation — scaling based on the CFG-like techniques —to validate this concept. **Whether other parallel architectures are effective does not affect the core contribution and conclusion of this paper.**
>
> - In fact, regarding alternative parallel architectures, we did explore several variations in Appendix B (Table 5), which showed minimal performance differences. This initial finding suggested that **the amount of parallel computation, rather than the specifics of the architecture, was the dominant factor.**
>
> - Furthermore, to directly test your insightful hypothesis of a more "tightly-coupled" design, we implemented a new variant that averages the K/V outputs at each layer before branching out again, based on 0.6B model pre-trained on StackV2 for 42B tokens, and reported the training loss.
>
>
> | P | ParScale | ParScale (shared KV-cache) |
> | --- | --- | --- |
> | 1 | 1.1518 | \- |
> | 2 | 1.1276 | 1.1282 |
> | 4 | 1.1145 | 1.1135 |
> | 8 | 1.1019 | 1.1034 |
>
> This new result strongly reinforces our central thesis: **it is the act of scaling parallel computation itself, not the specific architectural implementation, that drives the performance gains.** We will add this valuable comparison to the paper (Appendix B, Table 5) to make our contribution clearer.
>
> > W2: Absence of a "Brute-Force" Upper Bound: The paper compares PARSCALE to parameter-scaled models with better inference efficiency. However, a crucial baseline is missing: a standard dense model that is P-times wider, which would have a roughly equivalent computational budget (FLOPs) to a PARSCALE model with P streams. While this "wide" model would be highly inefficient due to its low compute-to-memory-access ratio, including it as a theoretical "upper bound" would be highly informative. It would help quantify the performance trade-off inherent in reusing parameters versus simply adding more, providing a clearer picture of how effectively PARSCALE closes the gap with a brute-force scaling approach for a given computational cost.
>
> This is an excellent point, and we are happy to clarify that our paper already includes this "brute-force" upper bound comparison. The baseline you suggested—a wider dense model—is exactly what we refer to as our "parameter scaling" baseline (explained in Line 887). Our scaling law analysis provides a direct answer to your question about the performance trade-off. We quantified this gap with the following capacity ratio:
>
> $$\frac{\text{Capacity of ParScale}}{\text{Capacity of brute-force upper bound}}=\frac{N(k\log P+1)}{NP}=\frac{k\log P+1}{P}$$
>
> > Q1: Clarification on "Similar Effects as Parameter Scaling": The paper claims that PARSCALE has "similar effects as parameter scaling." I would appreciate clarification on this point. Does this mean that for an equivalent increase in total computation (FLOPs), scaling P yields a comparable performance improvement to scaling parameters? Or does it primarily mean that the performance improvements from scaling P can be modeled by a scaling law with a similar mathematical form (i.e., a power law) to that of parameter scaling?
>
> - Thank you for your question. We would like to clarify that your second understanding is correct: scaling P is similar to scaling the model parameter by $O(\log(P))$, as modeled by the scaling law.
>
> > Q2: Optimizing KV Cache Memory Efficiency: The PARSCALE approach can be viewed as increasing the effective batch size of computation over a shared set of parameters, which successfully improves the compute-to-memory-access ratio for the model weights. However, as implemented, the KV cache size scales linearly with P, which could become the next memory bottleneck, especially for long contexts. Have the authors considered or explored strategies to optimize this? For example, could the P streams potentially share or contribute to a common KV cache, particularly for the initial shared context tokens, to further improve memory efficiency?
> - Thank you for your suggestion. As mentioned above (in our response to W1), we conducted this experiment, where the KV cache is averaged and shared across different streams. The results show that ParScale can further reduce KV cache usage while maintaining similar performance.
> ---
> Thank you again for your invaluable feedback. We hope our response could effectively address your concerns. Given the extend of our evaluation, we would be grateful if you would reconsider your evaluation of our manuscript.

---

> > ### Comment · Area_Chair_TWv6 · 2025-08-05
> > **Discussion**
> >
> > Dear Reviewer 8hYJ,
> >
> > The authors have responded to your concerns. How does their response change your view of the paper? If it does not, please clarify what the authors can do to address your concerns. If it does, please consider adjusting your score based on their response.
> >
> > Your AC

---

### Official Review · Reviewer_Qbt2 · 2025-07-12

**Clarity:** 3
**Significance:** 3
**Originality:** 2
**Rating:** 4
**Confidence:** 3

**Summary:**

This paper proposes ParScale, a scaling strategy that improves language-model performance by re-using the same parameters for P parallel forward passes on differently-transformed inputs via learnable prefixes and then aggregating the outputs with a lightweight MLP. Extensive pre-training experiments (~10^6 GPU-h) on 42B and 1T tokens confirm a log-linear scaling law: P-way parallel computation yields the same loss reduction as multiplying parameter count by O(log P), while adding only ~0.2 % parameters and incurring modest memory/latency overhead. A two-stage training recipe, i.e., pre-train normally on 1T tokens, then PARSCALE-fine-tune on 20B (2%), eliminates most of the extra cost and allows off-the-shelf models (e.g., Qwen-2.5) to be retro-fitted via parameter-efficient fine-tuning, supporting dynamic P-switching at inference.

This work provides both theoretical grounding and compelling empirical evidence that parallel computation can substitute for parameters, especially on reasoning-heavy benchmarks (code/math).
Compared with dense scaling, MoE, or inference-time scaling, PARSCALE is architecture-agnostic, training-time compatible, and memory-efficient, which is highly attractive for edge deployment.

**Questions:**

1. Could you elaborate more on the details of how to conduct prefix tuning as input transformation.
2. In Table 4, can the authors add the actual number of FLOPs for the baseline and ParScale, I'm curious about the results under fair FLOPs comparisons.

**Ethical Concerns:**

["NO or VERY MINOR ethics concerns only"]

**Limitations:**

yes

**Quality:**

3

**Strengths And Weaknesses:**

**Strengths**

* Strong empirical grounding, with 10^6 GPU-h scaling-law experiments yielding an R² ≈ 0.998 fit that cleanly isolates the effect of parallel computation versus parameters.
* The proposed two-stage training (1 T normal + ParScale) can still give large downstream gains.
* ParScale matches a 4.4B dense model with 22× less memory and 6× less latency at batch 1, which could potentially address resource-constrained deployment.

**Weaknesses**

* Even with the two-stage recipe, ParScale multiplies FLOPs by P during fine-tuning and still doubles total compute for P=8; this burden grows linearly with P and could outweigh benefits at larger scales.
* All latency/memory claims assume batch ≤8; at server-scale batch sizes the memory bottleneck evaporates and ParScale's compute overhead dominates, possibly erasing its edge over dense scaling.

---

> ### Author Rebuttal · Authors · 2025-07-31
>
> Thank you for your thorough and insightful review of our manuscript. We are grateful for your positive feedback, particularly recognizing our rigorous empirical grounding with large-scale scaling-law experiments and the significant efficiency gains of parallel scaling. Below are our responses to your concerns:
>
> > W1: Even with the two-stage recipe, ParScale multiplies FLOPs by P during fine-tuning and still doubles total compute for P=8; this burden grows linearly with P and could outweigh benefits at larger scales.
>
> We agree that the ParScale fine-tuning stage introduces a computational overhead. We see this as a strategic trade-off, and we would like to clarify that this concern is also applicable to other established scaling strategies. For example,
>
> - Parameter scaling increases FLOPs proportionally with model size, impacting the entire training pipeline and inference.
> - Inference-time scaling (e.g., scaling output tokens) also linearly increases FLOPs, often through less efficient serial computation.
>
> In contrast, ParScale strategically localizes the computational increase to the second, shorter fine-tuning stage. This one-time investment unlocks significant and persistent efficiency gains during inference (e.g., superior memory and latency). By shifting the computational burden away from inference and the initial pre-training phase, ParScale offers a more targeted approach to achieving the capabilities of larger models.
>
> > W2: All latency/memory claims assume batch ≤8; at server-scale batch sizes the memory bottleneck evaporates and ParScale's compute overhead dominates, possibly erasing its edge over dense scaling.
>
> -  Thanks for pointing it out. We respectfully suggest that **ParScale's value is best understood by its targeted strengths in low-resource environments and its adaptability, rather than judging it solely on its performance in high-throughput, large-batch scenarios.** Our work positions ParScale primarily as a solution for memory-limited and small-batch applications (like *edge computing*), where it demonstrably excels in both latency and memory efficiency.
> - Additionally, ParScale is compatible with Parameter-Efficient Fine-Tuning (PEFT) methods. This opens up possibilities for dynamic resource allocation in *server environments*. For instance, a model could operate in a low-FLOPs, low-cost mode (i.e., low P) during peak demand and scale up its computational budget (i.e., large P) with ParScale during off-peak hours for more powerful processing.
>
>
> > Q1: Could you elaborate more on the details of how to conduct prefix tuning as input transformation.
>
> - A detailed description of our prefix tuning method is provided in Appendix A. To be specific, we first duplicate the input x into P parallel copies, distinguishing them with different prefixes in each attention layer. This can be implemented as using different KV caches for different streams. We found that randomly initializing the prefixes is sufficient to ensure diverse outputs across different streams. We also leverage the prefix reparameterization trick.
>
>
> > Q2: In Table 4, can the authors add the actual number of FLOPs for the baseline and ParScale, I'm curious about the results under fair FLOPs comparisons.
>
> - Thank you for your suggestion. Assuming the input sequence length is S, as P increases, the FLOPs are 3.14S (P=1), 6.28S (P=2), 12.56S (P=4), and 25.12S (P=8), respectively, in units of GFLOPs.
> - However, we wish to clarify that our primary goal here is not a direct FLOP-for-FLOP "fair" comparison. Instead, the experiment is designed to demonstrate a new scaling dimension: showing that for a *fixed parameter budget*, model performance scales directly with increased *parallel computation*. **Even with larger FLOPs, ParScale could achieve lower memory consumption and latency, shown in our cost analysis.**
>
> ---
>
> Thank you again for your invaluable feedback. We hope our response could effectively address your concerns, and we would be grateful if you would reconsider your evaluation of our manuscript.

---

### Note · Authors · 2025-08-12

We sincerely thank the Area Chairs and all reviewers for their valuable time and constructive feedback. Our work introduces a third, orthogonal dimension for efficiently scaling model capacity, and its core strengths—as widely recognized by the reviewers—are threefold:

- First, we propose an innovative and theoretically sound Parallel Scaling Law, moving beyond traditional space-intensive Parameter Scaling and time-intensive Inference-Time Scaling. This approach addresses the critical communication bottleneck in modern LLM systems (Reviewers *Qbt2, UM4q, T6Gx*).

- Our conclusions are substantiated by comprehensive and rigorous experiments, praised for their clarity, scale, and compelling results across diverse settings (Reviewers *Qbt2, 8hYJ, UM4q, T6Gx*).

- Our method yields significant practical efficiency gains, drastically reducing memory and latency for resource-constrained deployment (Reviewers *Qbt2, 8hYJ, T6Gx*).

We are particularly encouraged by the strong endorsements from Reviewers *UM4q* (`a strong paper`) and *T6Gx* (`a very solid paper`). We also respectfully note that the reviewers who initially provided borderline scores (4) did not engage in the discussion phase. We believe our rebuttal has adequately addressed their initial concerns (e.g., strategic compute trade-offs and architectural invariance).

Furthermore, we hope the impact of our findings could extend beyond LLMs. For example, our work provides a fundamental principle that explains techniques like CFG in diffusion models. We hope our paper could encourage the machine learning community to evolve the question from **“How many parameters does a model need?”** to **“How many parameters and how much computation does a model need?”**, and inspire more efficient scaling paradigms to scale the model capacity.

---

### Decision · Program_Chairs · 2025-09-17

**Decision:**

Accept (poster)

**Comment:**

All reviewers agreed this paper should be accepted: it contains a comprehensive suite of experiments, the proposed method is elegant, and the results are impressive. The main concerns were around framing: e.g., reviewer Qbt2 worried that all experiments were in a regime with a small batch size (≤ 8). The authors responded by clarifying that they are targeting low-resource scenarios. This an important, often unexplored regime for language models. Given this and the author responses to other reviewer concerns, this paper is a clear accept.

One aspect that did not come up in the discussion but I would urge the authors to consider is the strength of the claim of power law discovery. Something that has not received enough attention in the LM power-law research community is that it is notoriously difficult to identify such power laws, see e.g.,

Clauset, Aaron, Cosma Rohilla Shalizi, and Mark EJ Newman. "Power-law distributions in empirical data." SIAM review 51.4 (2009): 661-703.

Using the Huber loss is a standard trick to mitigate the sensitivity of the squared loss to outliers, however it is not at all obvious that this fitting procedure identifies a ground-truth power law. I encourage the authors to take this into account when making changes to the manuscript for the camera-ready version. Thank you!